# Conserved roles for the dynein intermediate chain and Ndel1 in assembly and activation of dynein

Kyoko Okada[1,4], Bharat R. Iyer [2,4], Lindsay G. Lammers[2], Pedro A. Gutierrez[1,3], Wenzhe Li [1], Steven M. Markus [2,5] & Richard J. McKenney [1,5]

Processive transport by the microtubule motor cytoplasmic dynein requires the regulated assembly of a dynein-dynactin-adapter complex. Interactions between dynein and dynactin were initially ascribed to the dynein inter-mediate chain N-terminus and the dynactin subunit p150[Glued]. However, recent cryo-EM structures have not resolved this interaction, questioning its impor-tance. The intermediate chain also interacts with Nde1/Ndel1, which compete with p150[Glued] for binding. We reveal that the intermediate chain N-terminus is a critical evolutionarily conserved hub that interacts with dynactin and Ndel1, the latter of which recruits LIS1 to drive complex assembly. In additon to revealing that the intermediate chain N-terminus is likely bound to p150[Glued] in active transport complexes, our data support a model whereby Ndel1-LIS1 must dissociate prior to LIS1 being handed off to dynein in temporally discrete steps. Our work reveals previously unknown steps in the dynein activation pathway, and provide insight into the integrated activities of LIS1/Ndel1 and dynactin/cargo-adapters.

Cytoplasmic dynein-1 (hereafter referred to as dynein) transports a wide variety of cellular cargoes toward the minus ends of micro-tubules. The importance of dynein's cellular functions is highlighted in patients with mutations in the genes that encode dynein or its reg-ulators. Dysregulated dynein function underlies several human neu-rological diseases, including the malformations in cortical development (MCD) diseases lissencephaly, polymicrogyria, and microcephaly[1–7]. Although recent biochemical and structural studies have begun to shed light on the molecular basis by which this enor-mous motor complex is assembled and activated to transport its myriad cellular cargoes[8,9], numerous questions remain unanswered. Primary among them are the precise roles of the ubiquitous regulators LIS1, and the paralogs Nde1/Ndel1 in the dynein pathway.

Like most molecular motors, dynein assumes an autoinhibited conformation that prevents movement in the absence of cargo[8,9]. The current model for dynein activation posits that the motor must first be released from this state, referred to as the phi particle (due to its resemblance to the Greek character)[10]. Upon adopting the uninhibited, but still non-motile, open conformation – with the help of the lissencephaly-related protein LIS1 – it becomes competent for inter-actions with various cargo-specific adapter molecules and its activat-ing complex dynactin[9,11]. Once assembled, super-processive dynein-dynactin-adapter (DDA) complexes transport cargos over long dis-tances along microtubules[12,13]. Cryo-electron microscopy (cryo-EM) studies have revealed that dynactin serves as a platform that can scaffold as many as two dynein homodimers with their motor domains aligned in a parallel manner that promotes microtubule binding and processive motility[14,15].

Cryo-EM studies have also shown, at least in part, how dynein interacts with dynactin. Specifically, the N-terminal tail domains of

[1]Department of Molecular and Cellular Biology, University of California, Davis, Davis, CA 95616, USA. [2]Department of Biochemistry and Molecular Biology, Colorado State University, Fort Collins, CO 80523, USA. [3]Present address: Department of Biological Sciences, Columbia University, New York, NY 10027, USA. [4]These authors contributed equally: Kyoko Okada, Bharat R. Iyer. [5]These authors jointly supervised this work: Steven M. Markus, Richard J. McKenney. ✉e-mail: Steven.Markus@colostate.edu; rjmckenney@ucdavis.edu

each dynein heavy chain (HC) dock onto dynactin's mini actin filament[16]. However, studies that predate identification of these HC-dynactin contacts revealed interactions between the dynein intermediate chain (IC; an accessory subunit of the dynein complex), and the p150[Glued] (hereafter p150) subunit of the dynactin complex[17,18]. Notably, the regions that mediate this interaction are absent from current cryo-EM structures of DDA complexes, raising questions about the physiological importance, and role of the IC-p150 contact in DDA complex assembly.

The absence of these regions from cryo-EM structures is likely due to the flexibility of the relevant portions of IC and p150: the IC N-terminus (ICN), and the C-terminal portion of the first coiled-coil of p150 (p150[CC1b])[17–22]. Within the dynactin complex, p150[CC1b] is part of an elongated structure that cryo-EM studies have revealed can exist in two possible states: (1) an autoinhibited docked state (minority of particles), in which p150[CC1] is anchored to the pointed end of the actin filament; and, (2) an undocked state (majority of particles), in which this region is not visible due to its flexibility[16,23–25]. In contrast to p150[CC1], the ICN is largely unstructured with a short single alpha helix (SAH) at the very N-terminus that makes direct contact with p150[CC1b] (refs. 19,20,26), and is also absent from all cryo-EM structures of DDA complexes. Therefore, despite a preponderance of biochemical evidence for interactions between these regions of dynein and dynactin, the field lacks a coherent model for what role(s) these contacts play in the formation of activated DDA complexes.

In addition to p150[CC1b], the ICN SAH also interacts with the dynein regulator Nde1[19,27]. The current model for Nde1 (and its paralog Ndel1) function posits that an ICN-bound Nde1 helps recruit LIS1 to the dynein motor domain[28–31]. However, one study reported a direct interaction between Ndel1 and the dynein motor domain[32], suggesting Nde1/Ndel1 may affect dynein function from two distinct dynein surfaces (the ICN, and the motor domain). In light of the shared binding region on the ICN for p150[CC1b] and Ndel1 – which compete for binding[19,27] – it is unclear at what point in the dynein activation cycle Ndel1 and p150[CC1b] bind to the ICN, and whether these interactions are required for DDA assembly, stability, or motility.

Here we set out to assess the roles of the ICN and Ndel1 in the function and assembly of dynein and DDA complexes. We performed assays in both budding yeast and mammalian systems to determine the extent of evolutionary conservation of these roles. Using a combination of approaches, we find that although the ICN is dispensable for dynein complex integrity, it is a conserved hub that mediates interactions with both Ndel1 and p150 that are required for the assembly and activity of DDA complexes in vitro and in vivo. We find that contacts between ICN and p150 are critically important during DDA assembly, and that they likely persist during processive motility. With the help of Alphafold2 (AF2)[33], we identify and validate residue-specific interaction surfaces between ICN, Ndel1 and LIS1, and surprisingly find overlapping binding sites on LIS1 for both Ndel1 and dynein, revealing that LIS1 must dissociate from Ndel1 prior to binding the HC. Our data provide new insight into the assembly and activation of DDA complexes and improve our understanding of Nde1's role in this process.

## Results

### Deletion of the ICN does not disrupt dynein complex integrity or motor activity

The dynein complex consists of two copies each of the heavy chain (HC, Dyn1 in yeast), the intermediate chain (IC, Pac11 in yeast), the light-intermediate chain (LIC, Dyn3 in yeast), and each of the light chains (LC; LC8, TcTEX, Robl in humans; Dyn2 in yeast). While the LCs are thought to help dimerize the ICs, the LICs are involved in mediating interactions with cargo adapter molecules[11]. However, the role of the ICs is less clear. These molecules consist of a disordered N-terminal region, a LC-binding region, and C-terminal WD repeats, which

assemble into a beta-propeller that associates with adjacent HC N-terminal tail domains[14,15] (Fig. 1a, b, and Supplementary Fig. 1a). The N-terminal regions of the ICs are largely unstructured, but possess one (yeast) or two (human) short alpha helices (SAHs) at the extreme N-termini (Fig. 1b, and Supplementary Fig. 1a, b)[20,34–37]. To understand the role of the ICN in dynein function, we generated mutant IC alleles of yeast and human dynein as follows: for yeast dynein, we deleted the N-terminal 43 amino acids of Pac11, the only IC variant in this organism; for human dynein, we removed the N-terminal 70 residues of IC2C (Fig. 1b, and Supplementary Fig. 1b), which is the most ubiquitously expressed isoform in humans[38,39]. These regions were selected based on their known contacts with p150[17,19,27], and do not include the adjacent sequences required for LC binding[39,40].

Wild-type (WT) and dynein[ΔICN] complexes were purified from yeast or human cells. Size-exclusion chromatography and mass photometry revealed no major differences in the shape or masses between WT and mutant dynein complexes, revealing that the ICN is not critical for dynein complex stability (Fig. 1c, d). Consistently, dynein[ΔICN] complexes were indistinguishable from WT motors when viewed by negative stain electron microscopy (Fig. 1e). To confirm the functionality of the mutant motors, we utilized either single molecule assays for yeast dynein, which moves processively on its own[41], or multi-motor microtubule gliding assays for human dynein which does not move processively in isolation[12,13,42]. Both yeast and mammalian dynein[ΔICN] complexes were motile, confirming the integrity of the mutant dynein motors (Fig. 1f). The only notable difference in motility parameters between WT and dynein[ΔICN] was a modest reduction in run length for the mutant yeast dynein complex. Finally, we confirmed that removal of the ICN did not disrupt binding of the LCs to the IC by SDS-PAGE for human dynein (Fig. 1c), or via dual-color single molecule assays, in which the yeast HC (Dyn1) and LC (Dyn2) were simultaneously visualized (Fig. 1g). For the latter, relative intensity values of fluorescently labeled Dyn2 revealed no significant difference between mutant and WT dynein complexes, indicating a similar degree of LC occupancy for each. We conclude that removal of the IC N-terminus does not disrupt dynein complex stability or activity.

### The ICN is critical for dynein function in vivo

We next wondered how deletion of the ICN would affect dynein function in cells. Dynein plays key roles in mitotic spindle assembly and positioning in many organisms[43–46]. To examine the role of the ICN in vivo, we generated HEK293 cells that inducibly express either WT or IC2C[ΔICN]. Immunofluorescence analysis of these cell lines revealed an approximately 3-fold increase in the mitotic index for cells expressing IC2C[ΔICN] compared to those expressing WT IC2C (Fig. 2a, b, Table 1). Closer inspection revealed that a large fraction of these cells showed aberrant multipolar or disorganized mitotic spindles, a hallmark of dynein dysfunction[46], likely accounting for the increased mitotic index. Thus, the ICN is required for proper spindle assembly and mitotic progression in mammalian cells, demonstrating a critical role for this domain in dynein function in vivo. In the budding yeast Saccharomyces cerevisiae, the only known function for dynein and dynactin is to position the mitotic spindle within the bud neck prior to anaphase onset[47–49]. Consistent with our observations in mammalian cells, deletion of the Pac11 ICN resulted in a spindle positioning defect as severe as deletion of the dynein heavy chain gene revealing an essential role for the ICN in yeast dynein function (Fig. 2c, Table 1). These observations demonstrate an evolutionarily conserved role for the ICN in dynein function in mammalian and yeast cells.

Processive dynein motility requires its association with activating cargo adapters and the dynactin complex[12,13]. Given the previously identified interaction between the ICN and the p150 subunit of dynactin, we wondered whether the ICN plays a role in the association between dynein and dynactin in the context of the activated DDA co-complex. To address this, we combined purified WT or

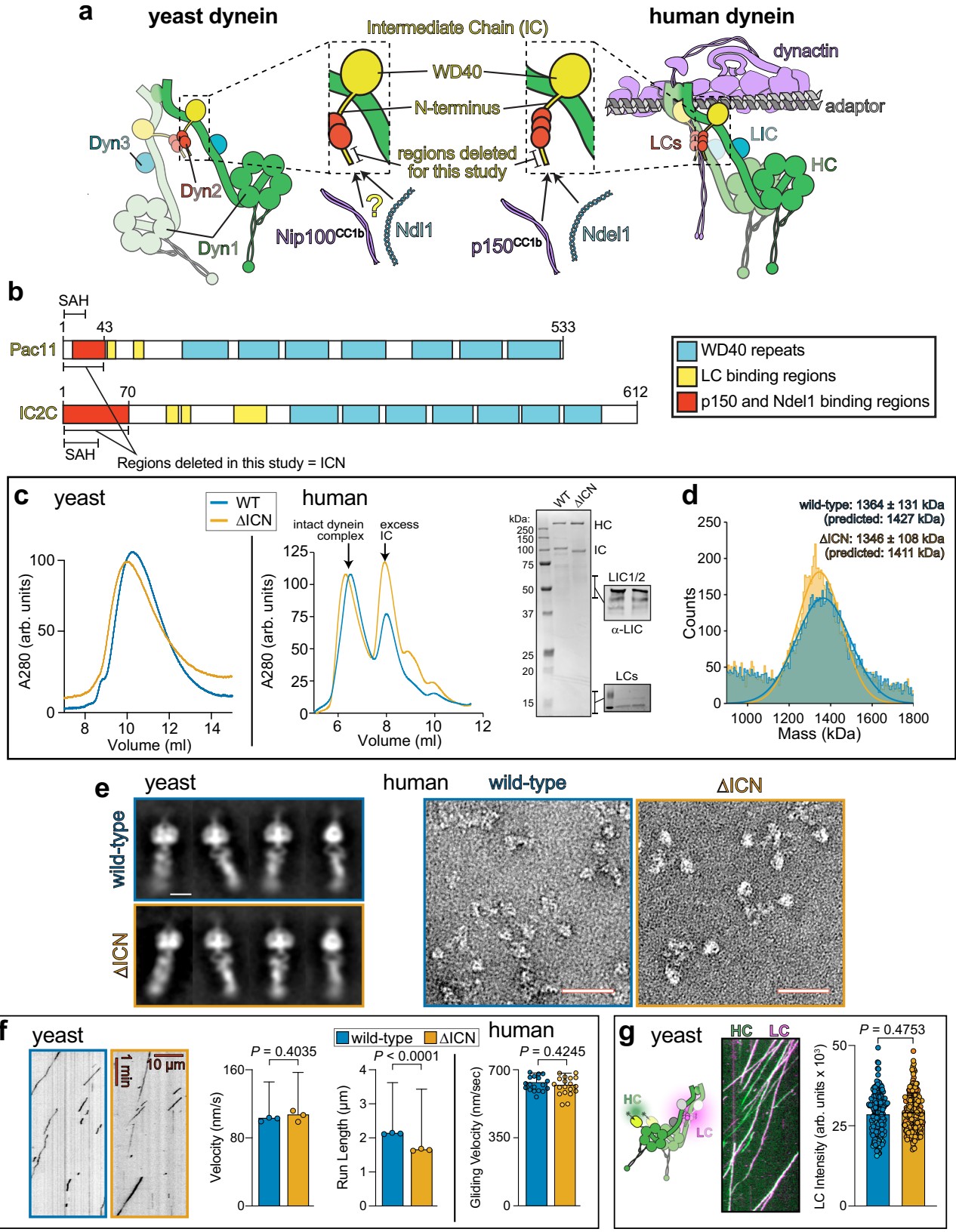

human dynein[ΔICN], mammalian dynactin, and the cargo adapter Hook3 (ref. 12), and assayed DDA complex formation via affinity isolation. Consistent with prior results[18,50–52], we observed an interaction between WT dynein and dynactin in the absence of Hook3, while no dynactin bound to dynein[ΔICN] (Fig. 2d). Addition of Hook3 led to formation of dynein·dynactin·Hook3 (DDH) complexes with WT dynein, but not with dynein[ΔICN] (Fig. 2d). We conclude that the interaction between the ICN and p150 is essential for the assembly of DDA complexes in vitro, and that the inability to form DDA complexes likely underlies the spindle assembly and positioning defects observed in IC2C[ΔICN]-expressing human cells, and Pac11[ΔICN]-expressing yeast cells.

**Fig. 1 | Deletion of the intermediate chain N-terminus has no effect on dynein complex integrity or function. a** Cartoon depicting the yeast and metazoan dynein complexes with (right) and without (left) the activating cargo adapter and dynactin. Inset depicts heavy chain- (HC) and light chain (LC)-bound intermediate chain (IC) with N-terminus (ICN) highlighted. Studies suggest that the ICN is an interaction site for both coiled-coil 1b (CC1b) of p150 (Nip100 in yeast) and Ndel1 (Ndl1 in yeast). **b** Schematics of ICs from budding yeast (Pac11) and humans (Dync1I2/IC2C) with domains indicated. SAH, single alpha-helix[20]. **c** Analytical size exclusion chromatography of wild-type (WT) and dynein$^{\Delta ICN}$ complexes purified from yeast and human cells, along with coomassie-stained gel and immunoblots of human dynein depicting the presence of accessory chains. Data from five independent replicates. **d** Mass photometric analysis of purified human dynein complexes. **e** Negative stain electron micrographs reveal intact dynein complexes from yeast (2D class averages shown; $n = 2$; scale bar, 10 nm) and human (raw images shown; $n = 2$; scale bars, 50 nm). **f** Representative kymographs, and plots (mean ± SD, along with mean values from individual replicates for single molecule data, with circles representing all data points for gliding velocity; mean run length values were determined by fitting raw data to one phase decay) depicting motility parameters for purified yeast (using single molecule assays; $n = 183/177/174$ WT, and $183/177/208$ $\Delta ICN$ motors from 3 independent replicates; $P$ values were calculated using a Mann–Whitney test) and human (via microtubule-gliding assays, $n = 20/20$ microtubules for WT and dynein$^{\Delta ICN}$ from 2 replicates; $P$ value was calculated using a two-tailed $t$-test) dynein reveal similar motility parameters between WT and dynein$^{\Delta ICN}$ complexes. **g** Kymograph depicting two-color single molecule motility assay in which yeast dynein HC (HaloTag$^{JFX49}$-Dyn1) and LC (Dyn2-S6$^{LD650}$) are visualized together. Plot (mean ± SD, along with all data points) depicts fluorescence intensity values for single molecules of Dyn2 bound to either WT or dynein$^{\Delta ICN}$, indicating the mutant binds to the same number of LCs as the WT complex ($n = 101/100$ and $101/100$ Dyn2 foci from WT and dynein$^{\Delta ICN}$ motors, respectively, from 2 independent replicates, represented by different shades of blue and orange). $P$ value was calculated using a Mann–Whitney test.

In budding yeast, pre-assembled dynein-dynactin complexes are delivered to cortical Num1 receptor sites from the plus ends of microtubules. Current data indicate that dynein-Pac1-Bik1 (homologs of human LIS1 and CLIP-170, respectively) complexes first associate with microtubule plus ends, which subsequently recruit dynactin prior to offloading to Num1[53–57]. These adapter-independent dynein-dynactin complexes are likely analogous to those observed above with metazoan proteins[18,50–52]. To assess the role of the ICN in dynein-dynactin binding in yeast, we imaged cells expressing Dyn1-3GFP and Jnm1-3mCherry (homolog of dynactin subunit p50/dynamitin) and quantified the extent of their localization to plus ends, the cell cortex, and spindle pole bodies (SPBs). Whereas WT cells exhibited Dyn1 and Jnm1 foci at all three sites to varying extents (Fig. 2e), the majority of which were colocalized (Fig. 2f), those expressing the Pac11$^{\Delta ICN}$ mutant exhibited very few colocalized foci, indicating that the ICN is required for dynein-dynactin binding at plus ends and the cell cortex. The lack of dynein and dynactin foci at the cell cortex in these cells (Fig. 2g, Table 1) is consistent with their co-dependence for Num1 binding. However, we also noted a reduction in plus end localization, suggesting that the ICN plays a role in dynein-Pac1 and/or Bik1 binding (see below).

To determine if the ICN of yeast dynein is required for interaction with the p150 homolog in this organism (Nip100), we employed a TIRF microscopy-based assay to measure direct binding between purified yeast dynein and the coiled-coil 1 domain of Nip100 (Nip100$^{CC1}$; Fig. 2h). Whereas WT dynein recruited Nip100$^{CC1}$ to microtubules, dynein$^{\Delta ICN}$ complexes did not (Fig. 2i), indicating the ICN indeed mediates the interaction between dynein and Nip100, consistent with prior results using mammalian proteins[19,27]. Taken together, these data indicate that the ICN plays an evolutionarily conserved role in both adapter-dependent and independent dynein-dynactin binding.

## ICN-bound Ndel1 recruits LIS1 to promote dynein localization and activity in cells

In addition to its interaction with p150, previous studies have found that the ICN is also the binding site for Ndel1, which has been implicated in promoting LIS1-dynein binding[28–31]. To determine if the ICN of yeast dynein is also required for this interaction, we employed our TIRF microscopy-based assay to measure binding between yeast dynein and Ndl1, the yeast homolog of Ndel1 (Fig. 3a). Whereas WT dynein strongly recruited Ndl1 to microtubules, dynein$^{\Delta ICN}$ complexes did not, indicating the ICN is indeed a conserved hub for binding Ndl1/Ndel1 and p150/Nip100 in yeast and mammals (Fig. 3b)[19,27]. These data also indicate that the ICN is the only contact point on the yeast dynein complex for Ndl1.

Although Ndl1/Ndel1 have been implicated in recruiting Pac1/LIS1 to dynein[28–31,58], the extent to which they do so in cells, and the roles of the ICN in these activities are unclear. Using TIRFM recruitment assays,

we found that both yeast and human dynein were able to bind Pac1/LIS1 in the absence of Ndl1/Ndel1. However, increasing concentrations of Ndl1/Ndel1 led to a significant increase in the extent of dynein-Pac1/LIS1 association (Fig. 3c, d; between a 3 and 5-fold increase). In contrast, we observed no such increase for yeast dynein$^{\Delta ICN}$-Pac1 binding, and a greatly attenuated enhancement for human dynein$^{\Delta ICN}$-LIS1 binding (Fig. 3c, d), indicating that ICN-bound Ndl1/Ndel1 is important for the Pac1/LIS1 recruitment to dynein. The extent of the recruitment of Pac1 by Ndl1 was biphasic, with peak recruitment (at 20 nM) followed by a decrease in Pac1 binding at higher Ndl1 concentrations. We noted similar trends for human dynein$^{\Delta ICN}$, as well as WT human dynein, albeit to lesser extents. We interpret this result as saturation of the microtubule-bound dynein by Ndl1/Ndel1, and subsequent sequestration of Pac1/LIS1 by the excess unbound Ndl1/Ndel1 in solution (see below).

Our data suggest that the role of ICN-bound Ndl1/Ndel1 is to promote dynein-Pac1/LIS1 binding in cells. To validate this model, we assessed the extent of dynein localization in cells with and without Ndl1. It is well established that the degree of dynein-Pac1 binding in yeast directly correlates with the extent of plus end and cortical association of dynein in cells[55,57]. Thus, deletion of Ndl1 would be expected to reduce the extent of dynein targeting to these sites, while its overexpression would increase this localization. Comparison of dynein localization in WT cells versus those lacking Ndl1 (Fig. 3e)[58] or overexpressing Ndl1 (Fig. 3f) indeed supports this model. These data help explain the reduced localization frequency of dynein$^{\Delta ICN}$ – which cannot bind Ndl1 – in cells (Fig. 2g). Consistent with an important role for Ndl1 in dynein function, a significant fraction of $ndl1\Delta$ cells exhibit a spindle positioning defect (Supplementary Fig. 2a). As a more sensitive readout for dynein function, we quantified dynein-mediated spindle movements (Supplementary Fig. 2b)[59,60]. This revealed that cells lacking Ndl1 indeed possess some dynein activity, albeit to a lesser extent than WT cells. In addition to lower dynein activity metrics (i.e., a reduced extent of dynein-mediated spindle movements, and a lower frequency of such events; Supplementary Fig. 2e, f), $ndl1\Delta$ cells exhibit a lower velocity of dynein-mediated spindle movements (Supplementary Fig. 2c) and a lower neck transit success frequency (Supplementary Fig. 2g), the latter of which is a read-out for dynein force production[60,61]. These data indicate that dynein does not absolutely rely on Ndl1 for its function (in contrast to Pac1[62]), but rather that Ndl1 promotes appropriate localization of dynein to its various subcellular locales by recruiting Pac1 to dynein, thereby enhancing its activity.

## Prediction and validation of a Ndel1-LIS1-ICN structure

Although it is well established that Ndl1/Ndel1 can simultaneously bind Pac1/LIS1 and the ICN (Fig. 3b, c)[30,63] structural information of this complex is lacking. We thus used AlphaFold2-Multimer (using Colabfold)[33,64] to generate models of the dimeric yeast and human

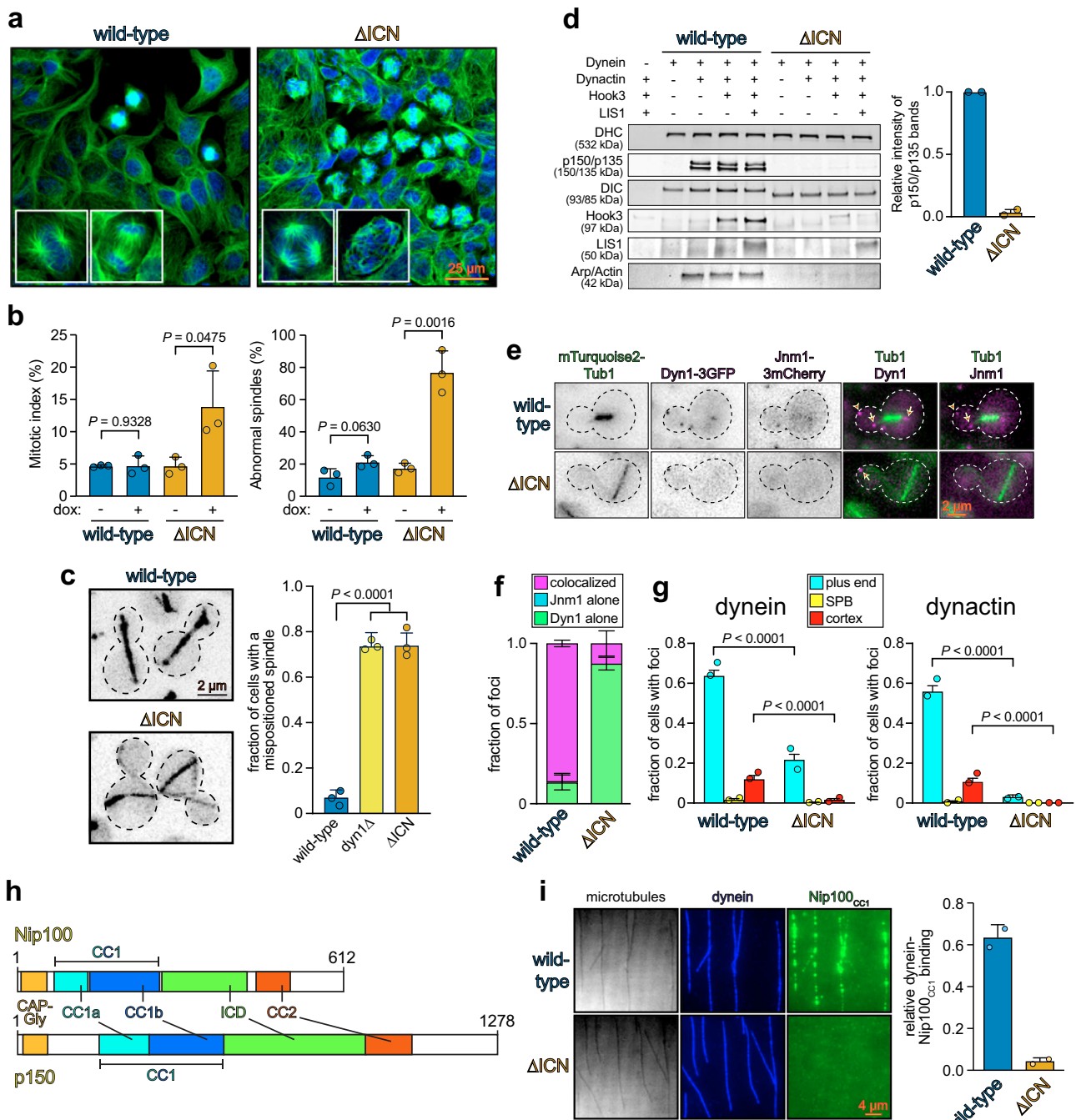

**Fig. 2 | The dynein intermediate chain N-terminus is required for in-cell dynein function and DDA assembly. a** Immunofluorescence images of Flp-In™ T-REx™ 293 cells inducibly expressing either WT or IC2C$^{\Delta ICN}$ ($n = 3$). **b** Plots (mean ± SD, along with mean values for individual replicates) depicting fractions of cells in mitosis (left), or those with abnormal spindles (right) for uninduced cells, or those induced to express WT or IC2C$^{\Delta ICN}$ ($n = 717/16/25$ uninduced WT cells, 802/50/27 induced WT cells, 1243/29/27 uninduced IC2C$^{\Delta ICN}$ cells, and 1150/45/33 induced IC2C$^{\Delta ICN}$ cells, from 3 independent replicates; $P$ values were calculated using a two-tailed $t$-test). **c** Representative images of cells expressing fluorescent α-tubulin and plot (weighted mean ± weighted standard error of proportion, along with values from individual replicates) depicting fractions of cells with mispositioned anaphase spindles ($n = 53/56/49$ WT cells, 38/56/61 $dyn1\Delta$ cells, and 49/66/69 $pac11^{\Delta ICN}$ cells, from 3 independent replicates). Two-tailed $P$ values were calculated from Z scores. **d** Immunoblots and plot (mean ± SD, $n = 2$) depicting relative degree of dynein-dynactin-Hook3 (DDH) assembly as a consequence of mutation or addition of factors. Quantification of the relative combined band intensities for p150/p135 in

the presence of dynein, dynactin, Hook3, and LIS1 is shown. $P$ value: two-tailed $t$-test. **e** Fluorescence images of cells expressing mTurquoise2-Tub1, Jnm1-3mCherry (homolog of human p50), and either WT or dynein$^{\Delta ICN}$–3GFP (arrows, plus end foci; arrowhead, cortical focus; $n = 2$). **f** Plot depicting degree of colocalization for indicated foci in WT or Pac11$^{\Delta ICN}$ cells (mean ± SD; $n = 221/193$ foci in WT cells, and 57/38 foci in $pac11^{\Delta ICN}$ cells from 2 independent replicates). **g** Plots (weighted mean ± weighted standard error of proportion, along with mean values from individual replicates) depicting fractions of cells exhibiting indicated foci in WT or $pac11^{\Delta ICN}$ cells (Dyn1: $n = 266/265$ WT cells and 208/200 $pac11^{\Delta ICN}$ cells from 2 independent replicates; Jnm1: $n = 266/265$ WT cells and 208/200 $pac11^{\Delta ICN}$ cells from 2 independent replicates). Two-tailed $P$ values were calculated from Z scores. **h** Schematics of human and yeast p150/Nip100 with domains indicated (CAP-Gly cytoskeleton-associated protein, glycine-rich, CC coiled-coil, ICD inter coiled domain). **i** Images and quantitation depicting degree to which Nip100$^{CC1}$ binds WT and dynein$^{\Delta ICN}$ (mean ± SD, along with mean values from individual replicates; $n = 20$ microtubules for each, from 2 independent replicates).

**Table 1 | Summary of expected and observed mutant phenotypes**

| | | Dynein mutants | | LIS1 mutants | Ndel1 mutants | |
| --- | --- | --- | --- | --- | --- | --- |
| Mutant: | | ΔICN | IC$^{AAA}$ | R316A/W340A or "5A" | Ndel1$^{AAA}$ | Ndel1$^{AA}$ |
| Expected outcome (defective in indicated activation steps; see Fig. 6): | | Lack of DDA assembly/doesn't bind Ndel1 (steps 2 and 4) | Inability to bind Ndel1 (step 2) | Inability to bind dynein and Ndel1 (prior to step 2, and step 3) | Inability to bind IC/ability to bind LIS1 (step 2) | Inability to bind LIS1/ability to bind IC (step 2) |
| Expected phenotype: | Yeast | Defects in spindle position, and localization of dynein and dynactin | No binding to Ndl1 in vitro | No binding to Ndl1 in vitro | No binding to IC in vitro/binding to Pac1 | No binding to Pac1 in vitro/binding to dynein complex |
| | Human | Defects in spindle assembly and DDA assembly | No binding to Ndel1 in vitro | No binding to Ndel1 in vitro | No binding to IC in vitro/binding to LIS1 | No binding to LIS1 in vitro/binding to IC |
| Outcome validated? | Yeast | Yes (Fig. 2c, e, g) | Yes (Supplementary Fig. 3c) | Yes (Fig. 4c) | Yes (Supplementary Fig. 3d, e) | Yes (Supplementary Fig. 3f, g) |
| | Human | Yes (Fig. 2a, b, d) | Not tested | Yes (Fig. 4d) | Yes (Wang and Zheng, 2011 JBC) | Yes (Wang and Zheng, 2011 JBC) |

Summary of expected and observed phenotypes for various mutant proteins examined in this study. The results from in vivo and in vitro assays are included. Note that steps refer to the model shown in Fig. 6.

versions of these complexes. The resulting AlphaFold2-Multimer (AF2) models for the human and yeast complexes (Supplementary Fig. 3a, b) are strikingly similar in their overall appearance, and in the relative positions of the predicted binding sites on Ndl1/Ndel1 for the ICNs and Pac1/LIS1. Whereas the majority of both Ndl1/Ndel1 proteins are comprised of coiled-coils, the C-termini of both are predicted to be unstructured, at least in the context of the Ndl1/Ndel1-Pac1/LIS1-ICN complex (Supplementary Fig. 3a, b). Pac1 and LIS1 are predicted to engage with Ndl1/Ndel1 using both of their WD40 beta-propeller domains, while short alpha-helices within the ICNs are predicted to make contacts with non-overlapping regions near the N-termini of Ndl1/Ndel1 (Fig. 4a, b, and Supplementary Fig. 3a, b). Close inspection of the human Ndel1-LIS1-ICN model revealed contact points that have been previously validated. In particular, Ndel1 E119 and R130 (Fig. 4a) have been shown to be important for LIS1 binding[65], while a group of glutamates at the N-terminus of Ndel1 are required for dynein binding (Supplementary Fig. 3a)[63]. NMR studies have also demonstrated the presence of a single alpha-helix (SAH) within the ICN that is required for Ndel1 binding[27]. Based on these experimental validations, we conclude that the AF2 models for the human Ndel1-LIS1-ICN complex is an accurate structural model.

To validate the AF2 model of the corresponding yeast proteins, we generated mutations within Ndl1 and the N-terminus of Pac11 that would be predicted to disrupt ICN-Ndl1 interactions (Supplementary Fig. 3b, interface 1), and Pac1-Ndl1 interactions (Supplementary Fig. 3b, interface 2), and employed our TIRFM-based assay to assess their effects on protein interactions. Mutating a cluster of positively charged residues at interface 1 in the ICN to alanines (Pac11$^{AAA}$) strongly disrupted ICN-Ndl1 binding (Supplementary Fig. 3c, Table 1), while mutation of negatively charged residues at the N-terminus of the Ndl1 coiled-coil domain (Ndl1$^{CC}$) to alanines (Ndl1$^{CC[AAA]}$) also significantly impaired the Ndl1-dynein interaction (Supplementary Fig. 3d, Table 1). Using mass photometry, we found that interface 1 mutations on Ndl1 had no effect on its interaction with Pac1 (Supplementary Fig. 3e, Table 1). Finally, mutation of E64 and R78 at interface 2 of Ndl1$^{CC}$ to alanines (analogous to E119 and R130 in Ndel1) disrupted Pac1 binding (Supplementary Fig. 3f) but not dynein binding (Supplementary Fig. 3g, Table 1). These results verify the structural organization of the ICN-Ndl1 and Pac1-Ndl1 complexes predicted by AF2 and indicate that they are largely conserved from yeast to human.

## Ndl1/Ndel1 competes with dynein for Pac1/LIS1 binding

Interestingly, the surface of the Pac1/LIS1 WD40 domain predicted to contact the Ndl1/Ndel1 coiled-coil overlaps with the reported dynein binding region of Pac1/LIS1[66,67]. This suggests that Pac1/LIS1 may only be able to bind to either Ndl1/Ndel1 or the dynein motor domain, but not both simultaneously. This observation contrasts with previous models that posited a tripartite Ndel1-LIS1-dynein motor complex[28–31]. Competition between Ndl1 and dynein for Pac1/LIS1 potentially accounts for the reduction in dynein-Pac1/LIS1 binding we noted in Fig. 3c, d at higher Ndl1/Ndel1 concentrations, which could be a consequence of excess Ndl1/Ndel1 in solution competitively binding to Pac1/LIS1.

To determine whether Pac1/LIS1 indeed employs the same surface to interact with Ndl1/Ndel1 and dynein, we assessed the ability of dynein-binding Pac1/LIS1 mutants[66,67] to interact with the coiled-coil regions of Ndl1 and Ndel1. We excluded the unstructured C-terminus of Ndel1/Ndl1 (see Supplementary Fig. 3a, b) to assess whether this region is required for dynein-binding, as has been previously suggested[32,68]. We mutated residues in Pac1 and LIS1 that have been shown to be important for dynein binding (R316A and W340A in LIS1; R275A, R301A, R378A, W419A, and K437A in Pac1; Fig. 4a, b), and employed mass photometry to assess complex formation. Whereas WT LIS1 and Pac1 both formed 1:1 dimeric complexes with Ndel1 and

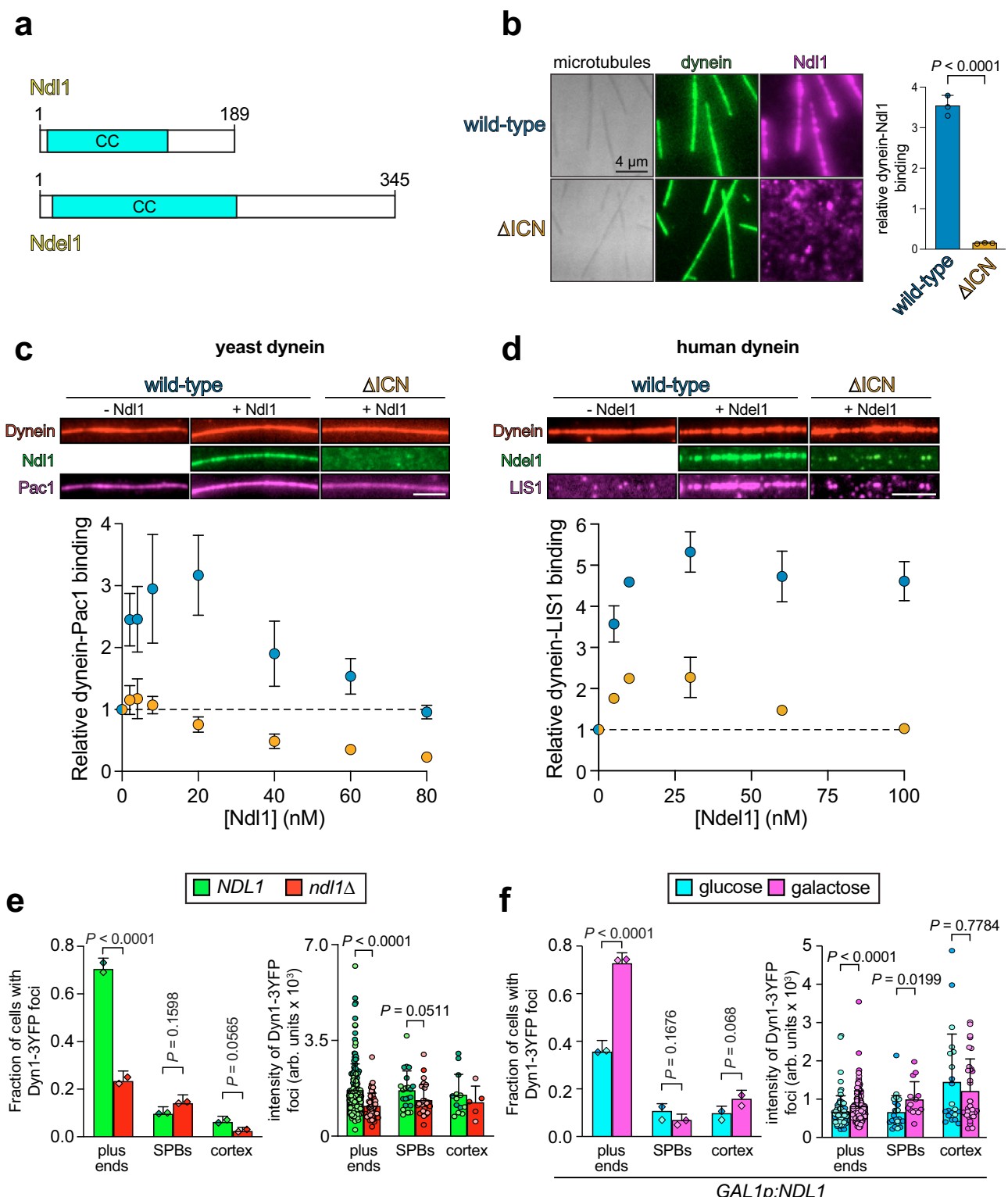

Ndl1 (Fig. 4c, d, Table 1), the extent of complex formation was either strongly reduced, or eliminated by the mutations. In further support of the notion that LIS1 uses the same binding interface for both Ndel1 and dynein, addition of Ndel1$^{CC}$ effectively prevented LIS1 from binding to a dimerized human dynein motor domain (dynein$^{MOTOR}$; which lacks the Ndel1-binding ICN; Fig. 4e) and to preassembled dynein-dynactin-Bicd2 (DDB) complexes (Fig. 4f). To determine if the same competition occurs in live cells, we assessed the ability of Ndl1 to compete Pac1 away from a yeast dynein motor domain fragment

(dynein$^{MOTOR}$) that is also lacking the ICN. Deletion of Ndl1 resulted in increased dynein$^{MOTOR}$-Pac1 binding (as apparent from the presence of plus end foci; Fig. 4g), while overexpression of Ndl1 had the opposite effect (i.e., reduced the frequency and intensity of dynein$^{MOTOR}$ foci; Fig. 4h). Taken together, these data indicate that the coiled-coil domain of Ndl1/Ndel1 and the dynein motor domain compete for Pac1/LIS1 binding, suggesting that Ndl1/Ndel1-recruited Pac1/LIS1 must first unbind from Ndl1/Ndel1 prior to binding dynein. Recently published data affirm this conclusion[69].

**Fig. 3 | ICN-bound Ndel1/Ndl1 recruits LIS1/Pac1 to promote in-cell dynein localization. a** Schematics of human Ndel1 and its budding yeast homolog Ndl1 with coiled-coil domains (CC) indicated. **b** Representative images and quantitation (mean ± SD, along with mean values from individual replicates) depicting role of ICN in dynein-Ndl1 binding ($n = 30$ microtubules for each, from 3 independent replicates). $P$ value: unpaired two-tailed Welch's $t$-test. Fluorescence images and quantitation (mean ± SD; dashed line indicates relative dynein-Pac1/LIS1 binding in the absence of Ndl1/Ndel1) depicting relative dynein-Pac1 (**c**) or LIS1 (**d**) binding. Binding was determined from relative intensity values for microtubule-associated Pac1 or LIS1 with respect to dynein (for **c**, 20 nM Pac1 and ~15 nM dynein were used; for **d**, 20 nM dynein and 60 nM LIS1 were used). Note that buffer conditions were used such that neither Pac1, LIS1, Ndl1, nor Ndel1 was recruited to microtubules in the absence of dynein. Thus, the degree of microtubule localization for Pac1, LIS1, Ndl1 and Ndel1 is directly proportional to the extent of their dynein binding (**c**:

$n = 10/10/10$ microtubules for WT, and $10/10/10$ microtubules for dynein$^{\Delta ICN}$, from 3 independent replicates; **d**: $n = 53/50$, 52/45, 56/56, 56/56, 52/72, 56/61 microtubules for WT, and 52/56, 58/53, 51/59, 54/71, 52/52, 55/57 microtubules for dynein$^{\Delta ICN}$, mean ± SD from 2 independent replicates; scale bars, 5 μm). **e**, **f** Plots (mean ± SD, as well as all data points for intensity values) depicting the extent of dynein localization in cells with and without Ndl1 (**e**: $n = 105/102$ $NDL1$ and $104/101$ $ndl1\Delta$ cells, and $95/95$ and $41/42$ foci from $NDL1$ and $ndl1\Delta$ cells, respectively, all from 2 independent replicates), or with and without overexpressed Ndl1 (**f**: $n = 113/100$ uninduced, and $102/100$ induced cells, and 74/52 and 103/106 foci from uninduced and induced cells, respectively, all from 2 independent replicates). $P$ values were calculated using a Mann–Whitney test (for intensity values), or by calculating Z scores (two-tailed). For cells in panel **f**, which were engineered to possess a $GAL1$ promoter upstream of the $NDL1$ locus, Ndl1 overexpression was controlled by the exclusion or inclusion of galactose in the media for 3 hours immediately prior to imaging.

## Competitive binding of p150$^{CC1}$ or Ndel1 to the ICN precludes DDA assembly

A well-established method to inhibit dynein function in cells involves microinjection or expression of p150$^{CC1}$ in mammalian cells[70]. Although this technique has been used in numerous studies, the mechanism by which this truncated protein precludes dynein function is unknown, as it does not disrupt dynactin integrity[71]. In light of our findings that this fragment makes direct contacts with the ICN, we hypothesized that p150$^{CC1}$ competes for ICN binding with the native p150 molecule in the dynactin complex. We first wondered whether expression of p150$^{CC1}$ also inhibits dynein function in budding yeast. To this end, we generated yeast strains engineered to conditionally overexpress Nip100$^{CC1}$ from the galactose-inducible $GAL1$ promoter ($GAL1p$). In contrast to uninduced cells, a large fraction of those overexpressing Nip100$^{CC1}$ possessed a mispositioned spindle, indicating that the capacity to disrupt dynein activity by this fragment is indeed conserved (Fig. 5a). We observed Nip100$^{CC1}$ colocalizing to the SPBs and to microtubule plus ends with dynein (Fig. 5b, arrowheads, and arrow, respectively), indicative of their binding in cells.

To directly test whether p150$^{CC1}$ competitively inhibits dynein-dynactin binding, we assessed the extent of DDA complex assembly in the absence or presence of this protein fragment. We also included recombinant LIS1 in our assays given its recently documented ability to promote DDA complex assembly[62,72–74]. Using the purified cargo adapter Hook3, we quantified the extent of DDH assembly in bovine brain lysate by affinity isolation (Fig. 5c). In the absence of p150$^{CC1}$, the addition of LIS1 promoted DDA assembly (by ~3-fold; Fig. 5d), consistent with recent reports. However, titration of increasing amounts of p150$^{CC1}$ significantly compromised this process, suggesting that p150$^{CC1}$ indeed competitively disrupts dynein-dynactin binding. We next wondered whether p150$^{CC1}$ could effectively compete with endogenous p150 for ICN binding after DDA complex assembly was already complete. To this end, we repeated our assay, but instead of including p150$^{CC1}$ coincident with LIS1 and Hook3, we added p150$^{CC1}$ 60 min after Hook3 addition. Interestingly, this led to almost no disruption in DDA assembly (Fig. 5d, hatched bar), suggesting that native p150 within the dynactin complex either exhibits higher affinity for dynein than the isolated p150$^{CC1}$ fragment, or that p150$^{CC1}$-ICN binding is not required to maintain assembled DDA complexes.

Previous studies have revealed that human p150$^{CC1}$ and Ndel1 compete for binding to the ICN[19,27], suggesting excess Ndel1 may also inhibit DDA assembly in vitro. Using our TIRFM-based binding assay, we found that yeast Nip100$^{CC1}$ and Ndl1 both exhibit similarly high affinity for the dynein complex (Supplementary Fig. 4a), and compete for binding to dynein (Supplementary Fig. 4b), much like their human counterparts. Moreover, we found that Ndl1 bound equally well to both WT yeast dynein, which exists predominantly in the phi conformation, and a mutant open dynein that cannot adopt the phi conformation (dynein$^{DK}$, Supplementary Fig. 4a)[62]. This is in contrast to

Pac1, which exhibits a greater degree of binding to the open mutant than WT, as expected (Supplementary Fig. 4c)[62].

In light of the in vivo data indicating an important role for Ndl1 in promoting dynein localization and function in yeast (Fig. 3d and Supplementary Fig. 2)[58], we wondered whether inclusion of Ndel1 in our DDA assembly assay would enhance LIS1 activity (i.e., increase DDA assembly), or whether its binding to the ICN would compete with p150 binding, and thus preclude complex assembly, similar to p150$^{CC1}$. To this end, we repeated our assay above (Fig. 5c) with increasing concentrations of Ndel1, which caused a dose-dependent decrease in DDA assembly (Fig. 5e). We noted that higher concentrations of Ndel1 were required to achieve similar degrees of complex disruption, suggesting that p150$^{CC1}$ may have higher affinity for ICN than Ndel1. Using our TIRFM-based binding assay, we found this to be the case, with p150$^{CC1}$ exhibiting a ~6-fold greater apparent affinity for dynein than Ndel1 (Supplementary Fig. 4d). Finally, as noted above for p150$^{CC1}$, addition of Ndel1 to lysates 60 min after addition of Hook3 did not disrupt DDA complex assembly (Fig. 5e, hatched bar). Thus, despite Ndel1's in-cell activity, which is to promote dynein-LIS1 binding, its excess in vitro perturbs DDA assembly in a manner similar to p150$^{CC1}$.

We wondered whether this inhibitory activity of Ndel1 is due to sequestration of either native or recombinant LIS1 in the lysates, thus preventing dynein-LIS1 binding. To test this, we repeated our DDA assembly assay using Ndel1 mutants that are unable to bind LIS1, but competent for interaction with dynein: E119A and R130A (see Fig. 4a, inset)[65]. We used the coiled-coil fragment of Ndel1, which is sufficient to interact with LIS1[75] (Fig. 4c). Inclusion of these mutants in the lysate concurrent with Hook3 prevented DDA assembly to the same extent as WT Ndel1$^{CC}$, indicating this inhibition is likely a consequence of Ndel1$^{CC}$-ICN binding, and not the result of LIS1 sequestration. Considering the similar degree of inhibition by Ndel1$^{CC}$ and full-length Ndel1 (compare Fig. 5e and 5f), these data further indicate that the C-terminus of Ndel1 is not required for its interaction with dynein.

Finally, we wondered why preassembled DDA complexes were refractory to Ndel1 and p150$^{CC1}$-mediated inhibition. We hypothesized that this was a consequence of one of two possible scenarios: (1) the p150$^{CC1}$-ICN contact is only required for assembly, but not for maintenance of assembled DDA complexes; or, (2) the ICN-p150$^{CC1}$ contacts are required for maintenance of the complex, but become sufficiently stabilized (possibly due to the avidity provided by HC-dynactin interactions) such that exogenous p150$^{CC1}$ or Ndel1 are no longer able to compete for binding. TIRFM imaging of single molecules of preassembled DDH complexes mixed with either fluorescent Ndel1 or p150$^{CC1}$ revealed that a very small proportion of motile DDH complexes colocalized with either Ndel1 or p150$^{CC1}$ across a five-fold range of concentrations (Fig. 5g, h). We interpret the lack of robust p150$^{CC1}$ or Ndel1 binding to motile DDH complexes as an indication that the ICN is stably bound to the p150 coiled-coil within the dynactin complex, and thus unable to interact with the exogenous polypeptides.

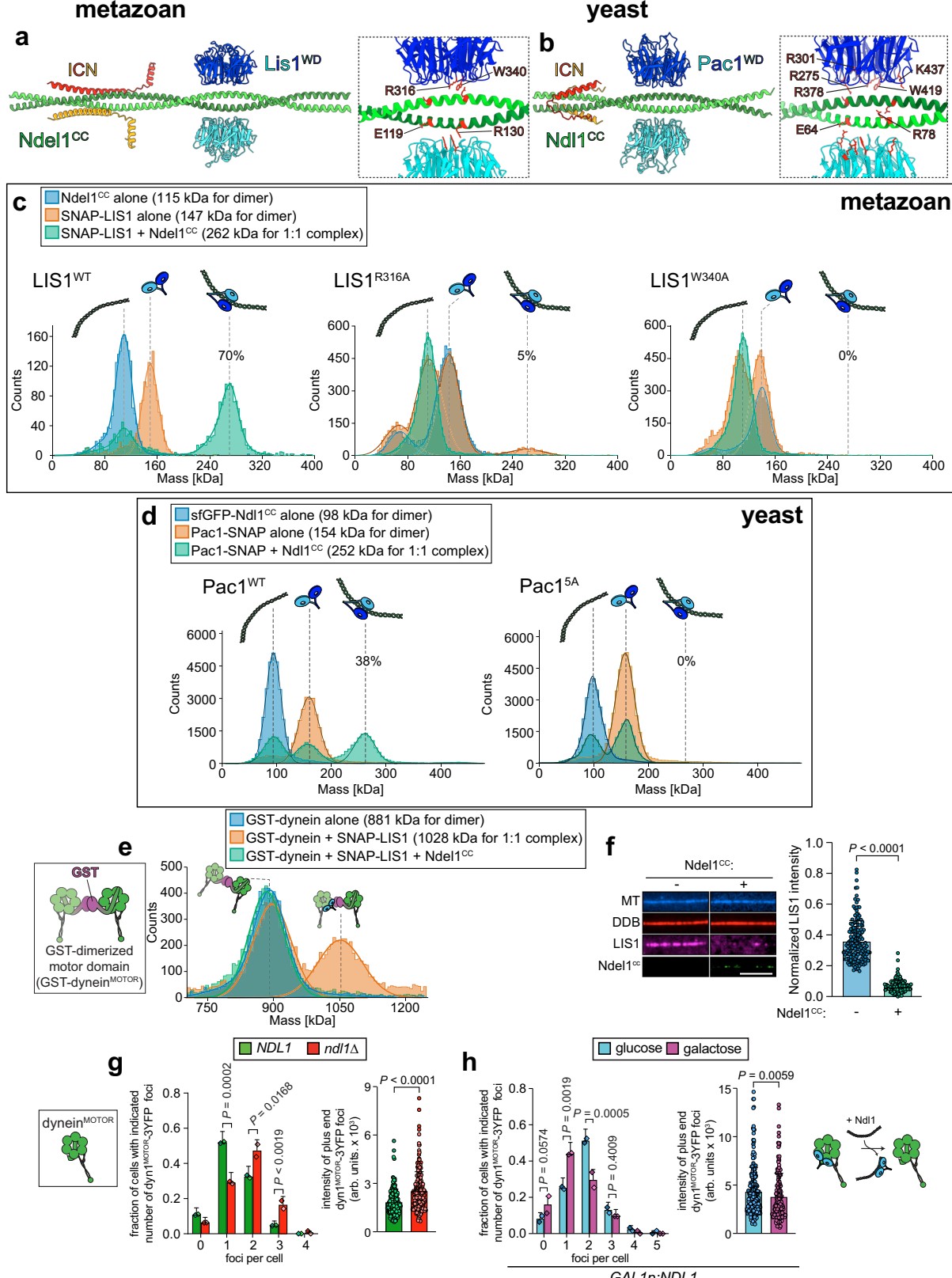

## Discussion

In this work, we find a remarkably limited region of the overall ~1.4 MDa cytoplasmic dynein complex is required for the activation of dynein motility by dynactin and cargo adapters both in vitro and in cells. While a direct interaction between the ICN and p150 was identified over 25 years ago[17], its role in dynein function has remained elusive, even after

the revelation of recent high resolution cryo-EM structures of assembled DDA complexes[14,16,24]. Our results reveal a conserved role for this interaction that spans over a billion years of evolution. In the absence of the ICN-p150 interaction, dynein is unable to interact with dynactin and cargo adapters, leading to behaviors that phenocopy the complete loss of dynein activity in cells. Our data explain long-standing

**Fig. 4 | Dynein and Ndel1/Ndl1 compete for binding to LIS1/Pac1. a, b** Alphafold2-Multimer models of 2 Ndel1$^{CC}$:2 LIS1$^{WD40}$:2 ICN (**a**) or 2 Ndl1$^{CC}$:2 Pac1$^{WD40}$:2 ICN (**b**) complexes (also see Supplementary Fig. S3). Insets highlight residues on each protein mutated in this study or others[63,66,67]. **c, d** Mass photometric analysis of individual proteins or mixtures thereof. Proteins were mixed together at a 1:1 molar ratio. Fits of mean mass values for each species, and relative fraction of particles with indicated mass are shown. Although the majority of species were comprised of 1:1 complexes, a minor but reproducible population of 2 Pac1:1 Ndl1 were apparent in all experiments with yeast proteins only (also see Supplementary Fig. S3e, f). **e** Mass photometry of the human GST-dimerized dynein motor domain (depicted in cartoon schematic) with and without LIS1 and Ndel1$^{CC}$. Proteins were mixed together at a 1:1 molar ratio. **f** Representative fluorescence images and quantitation depicting relative binding between microtubule-bound dynein-dynactin-BicD2 complexes (DDB) and LIS1 ± Ndel1$^{CC}$ in the presence of AMPPNP (mean ± SD, $n$ = 88/

71 microtubules for chambers without Ndel1$^{CC}$, and 57/68 microtubules for those with Ndel1$^{CC}$, from 2 independent replicates; $P$ value calculated using a one-way ANOVA; scale bar, 5 μm). **g, h** Plots (mean ± SD, as well as all data points for intensity values) depicting the extent of dynein$^{MOTOR}$ localization in cells with and without Ndl1 (**g**; $n$ = 66/63 *NDL1* and 64/63 *ndl1Δ* cells, and 87/82 and 115/107 foci from *NDL1* and *ndl1Δ* cells, respectively, all from 2 independent replicates), or with and without overexpressed Ndl1 (**h**; $n$ = 61/62 uninduced, and 62/63 induced cells, and 109/113 and 92/78 foci from uninduced and induced cells, respectively, all from 2 independent replicates). $P$ values were calculated using a Mann–Whitney test, or by calculating Z scores (two-tailed). For cells in panel **h**, which were engineered to possess a *GAL1* promoter upstream of the *NDL1* locus, Ndl1 overexpression was controlled by the exclusion or inclusion of galactose in the media for 3 h immediately prior to imaging. Cartoon schematic in **h** depicts the proposed manner by which Ndl1 competes Pac1 away from dynein$^{MOTOR}$.

observations in the dynein field. First, the mechanism of p150$^{CC1}$ as an inhibitor of dynein function in cells can be explained by our finding of a competitive interaction with the endogenous p150 within the dynactin complex for the dynein ICN. Second, excess Ndel1 disrupts dynein function in mitotic spindle assembly[31] and vesicular trafficking[68], likely because it competes with p150 during DDA assembly. Third, monoclonal antibodies that recognize the ICN[19] disrupt dynein transport in vivo[76–78], likely through perturbation of DDA assembly as shown here for both Ndel1 and p150$^{CC1}$, which also bind to the ICN. Finally, our data explain why Ndel1 acts as a positive modulator of dynein activity in both genetic and cell biological experiments[46,58,79–81], via its role in the direct recruitment of the dynein activator, LIS1, to the motor.

Despite much attention in the field, the precise roles of the ubiquitous and critical dynein regulators LIS1 and Ndel1 have remained stubbornly opaque[9,82]. The relatively recent discoveries of the dynein autoinhibition and activation pathways[9,11] have provided new context for possible roles of these molecules in dynein function. Recent work by several groups revealed LIS1's role in biasing the formation of activated DDA complexes by interfering with the autoinhibited phi conformation of the dynein dimer via its interactions with the dynein motor domain[62,72–74]. LIS1 binding is thought to stabilize the open conformation of the motor by acting as a check valve that prevents reversion of dynein back to the autoinhibited phi conformation, thus priming it for assembly with dynactin and cargo adapters[9]. However, the function of Ndel1 in this dynein activation process has remained unclear. Our current data confirm prior results suggesting a role for Ndel1 in the recruitment of LIS1 to the dynein motor via Ndel1's interaction with the ICN (Fig. 6). Our biochemical and cell biological data fit well with previous data suggesting that Ndel1 promotes LIS1 function in a range of model systems, from fungi to metazoa[58,82–84]. For example, it has been noted by several groups that Ndel1 depletion (or deletion) phenotypes can be rescued by LIS1 overexpression. Interestingly, the converse has also been noted: that Ndel1 overexpression can rescue LIS1 depletion – but not deletion – phenotypes[84,85]. These latter data are likely also explained by Ndel1 recruitment of LIS1 to dynein, thereby increasing the effective LIS1 concentration with respect to dynein. As haploinsufficiency of LIS1 causes lissencephaly, the cellular concentration of LIS1 is critical for human health. Thus, the recruitment of LIS1 to dynein by Ndel1 likely plays a key role in ensuring proper dynein activity and cellular homeostasis.

Prior models posited that Ndel1 recruits LIS1 to dynein through simultaneous binding of LIS1 to both Ndel1 and the dynein motor domain[28,30,31]. However, our new data challenges this notion, and instead supports a model in which LIS1 must unbind from Ndel1 prior to binding to dynein. In addition to scaffolding dynein and LIS1 within a single protein complex, this model suggests that Ndel1 may also function to prevent LIS1 from binding directly to the dynein motor domain in a temporally discrete step that precedes LIS1-induced dynein activation, as recently proposed by Garrott and colleagues (Fig. 6, steps 2 and 3)[82]. In this model, a yet unknown mechanism may

trigger a hand-off of LIS1 from Ndel1 onto the dynein motor domain, thus initiating LIS1-induced activation of DDA assembly. One such trigger may be the switch of dynein from its phi state – to which LIS1 cannot bind– to its open state to which LIS1 binds well (see Supplementary Fig. 4c)[62,72]. Alternatively, competitive binding of p150 to the ICN, which causes Ndel1-ICN unbinding, could potentially also disrupt Ndel1-LIS1 binding, facilitating the hand-off of LIS1 to the motor domain (Fig. 6, step 3). Post-translational modifications likely also play a role. Ndel1 in particular contains many phosphorylation sites[82], with one study showing weakened LIS1 binding as a consequence[86]. LIS1 is also a phosphoprotein and it is therefore conceivable that transition steps in our model (Fig. 6) are in part coordinated through changes in the phosphorylation status of Ndel1 and LIS1. Indeed, a recent report demonstrated that phosphorylation mimicking mutations of Ndel1 enhance its affinity for dynein, leading to enhanced disruption of DDA complexes in vitro[69].

What role does the ICN-p150 interaction play in DDA assembly? Cryo-EM and cross-linking coupled with mass spectrometry data have revealed that the elongated p150 projection arm folds back onto dynactin's actin-related protein 1 (Arp1) filament to interact with the pointed end of the dynactin complex[16,25] (see Fig. 6). In this conformation, the p150 arm occludes the interaction of cargo adapter proteins with the Arp1 filament, resulting in an autoinhibited dynactin (Fig. 6, step 3)[25]. Contacts have also been identified between p150$^{CC1a}$ and p150$^{CC1b}$ (ref. 21). Although unclear, this latter interaction may also play a role in maintaining the autoinhibited conformation of dynactin (see Fig. 6 inset with CC1a/1b). This model is consistent with single molecule observations that purified dynactin does not robustly interact with microtubules, whereas isolated p150 does[87]. Therefore, we speculate that the interaction between ICN and p150$^{CC1b}$ may represent the initial contact between dynein and dynactin that stimulates the release of the autoinhibited conformation of the p150 projection arm (Fig. 6, step 4), priming assembly of the fully active DDA complex (Fig. 6, step 5). However, our data also revealed that exogenous Ndel1 or p150$^{CC1}$ do not disrupt the integrity of preassembled DDH complexes, and do not robustly associate with processively moving DDH complexes. These data suggest that within the context of the fully assembled DDA complex, the ICN may be stably bound to p150 within the dynactin complex during dynein motility. Therefore, we conclude that the ICN-p150 interaction is not only required for initiation of DDA assembly, but is also a sustained contact within the fully assembled DDA complex.

We propose the following model to incorporate our data into the existing understanding of dynein activation. Dynein exists in equilibrium between the autoinhibited phi and open conformations (Fig. 6, step 1). Although we find that Ndl1 binds equally well to both conformations in vitro (Supplementary Fig. 4a), it remains to be determined if Ndl1/Ndel1 affects the equilibrium between these two conformations. Binding of Ndel1 to the SAH of the ICN (Supplementary Fig. 1b)[27,35] recruits LIS1 to dynein, while simultaneously preventing LIS1

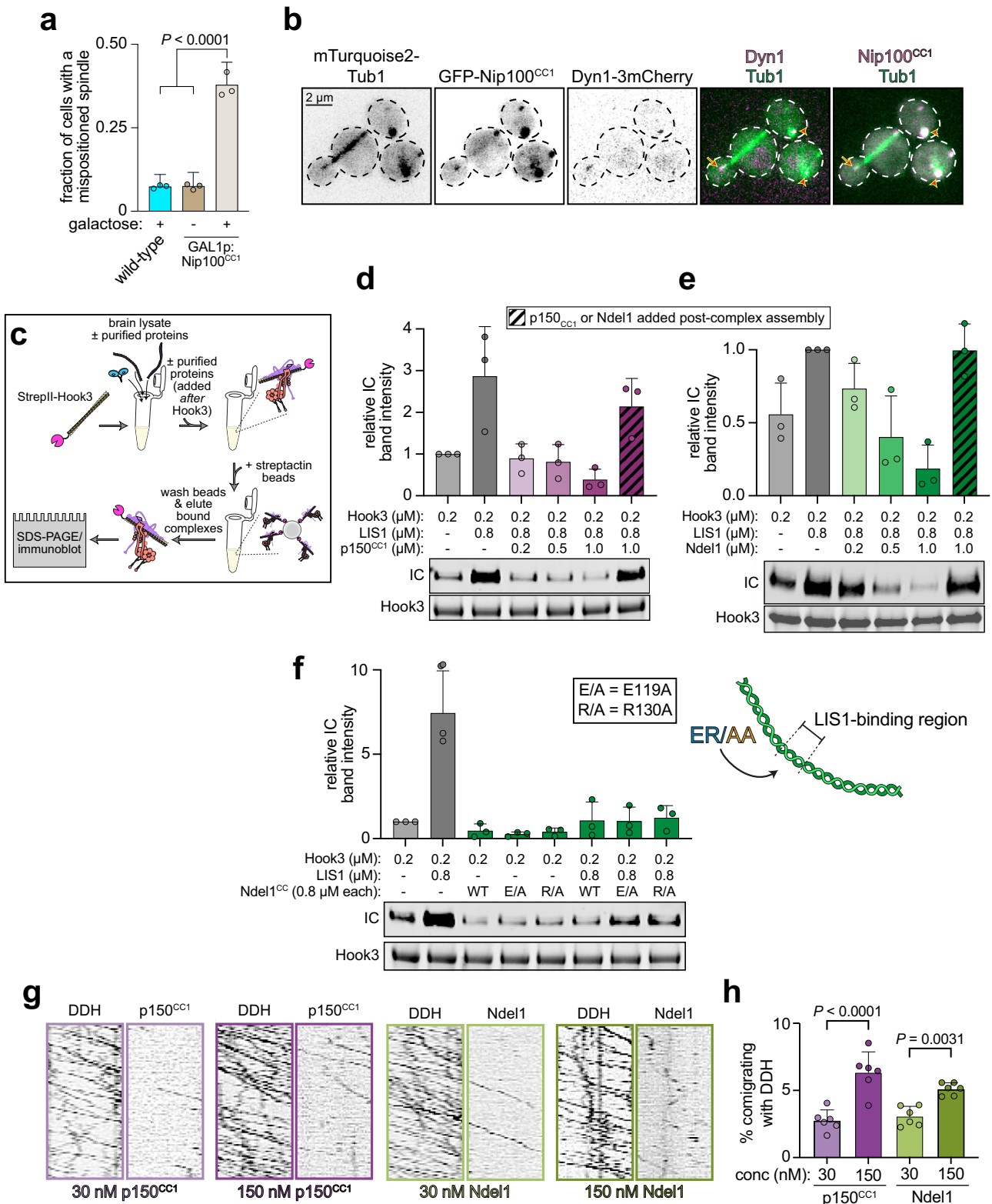

engagement with the dynein motor domain (Fig. 6, step 2). An unknown mechanism triggers the dissociation of LIS1 from Ndel1, and its subsequent interaction with the open form of the dynein motor, possibly concurrent with the dissociation of Ndel1 from the ICN. We propose that the interaction of the SAH and H2 of the ICN (Supplementary Fig. 1b) with dynactin-bound p150[CC1b] (refs. 20,27,35,37) relieves dynactin autoinhibition by preventing CC1 from contacting the dynactin pointed end, or by potentially interfering with the CC1a-

CC1b interaction[21] (Fig. 6, step 4). The resulting open dynein-LIS1-dynactin complex is not capable of processive motility, but is primed for interaction with a cargo-adapter molecule. This adapter-independent dynein-dynactin-LIS1 complex may represent the plus-end bound dynein complexes observed in yeast and metazoans[51–53,56,88]. This complex binds to one of the growing number of dynein cargo-adapters[11], which themselves are bound to various cellular cargos, leading to the formation of the active DDA complex that is competent

**Fig. 5 | Excess p150[CCI] and Ndel1 competitively inhibit DDA assembly but do not perturb pre-assembled complexes. a** The fraction of cells with a mispositioned spindle are plotted (weighted mean ± weighted standard error of proportion, along with values from individual replicates; $n = 53/64/44$ WT + galactose cells, 26/62/31 *GAL1p:NIP100[CCI]* + glucose cells, and 50/60/43 *GAL1p:NIP100[CCI]* + galactose cells, all from 3 independent replicates). Two-tailed *P* values were generated by calculating Z scores. **b** Representative fluorescence images of Nip100[CCI]-overexpressing cells (after growth in galactose-containing media) depicting localization of Nip100[CCI] within cells (arrows, plus end foci; arrowheads, SPB foci). **c** Cartoon schematic depicting experimental strategy to assess DDA complex formation in the absence or presence of indicated recombinant proteins, added either prior to addition of Hook3 to lysates, or 60 min thereafter. **d–f** Immunoblots and plots depicting

relative degree of DDH assembly as a consequence of addition of indicated wild-type factors (**d** and **e**), or indicated mutant Ndel1[CC] (**f**). **g** Representative kymographs from two-color movies depicting motility of 30 nM DDH complexes (assembled as indicated in panel **c**) in the presence of indicated concentration of fluorescent p150[CCI] or Ndel1. Note the low frequency of comigration of p150[CCI] and Ndel1 with DDH. **h** Plot (mean ± SD, along with values from independent replicates) depicting frequency of comigration of p150[CCI] or Ndel1 with DDH complexes ($n = 646/837/698/539/542/243$, 512/1020/449/646/670/558, 775/664/978/486/412/516, 2188/860/770/328/537/349 processively migrating DDH particles for 30 nM and 150 nM p150[CCI], 30 nM and 150 nM Ndel1, respectively). *P* values were calculated using a one-way ANOVA.

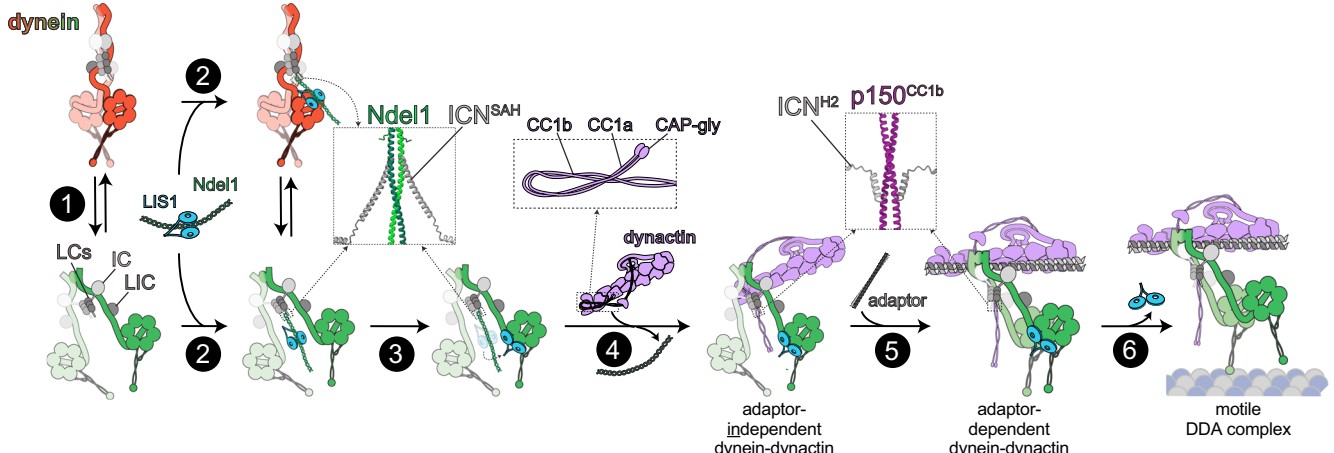

**Fig. 6 | Model for dynein activation. 1** Dynein stochastically switches between the autoinhibited phi and open states. **2** Ndel1 binding to the ICN SAH[27], which binds equally well to the phi and open states (see Supplementary Fig. 4a), recruits LIS1 to dynein. **3** An unknown mechanism leads to LIS1 unbinding from Ndel1, and subsequently binding to open dynein, thus stabilizing this conformation. **4** Interactions between H2 of the ICN and p150[CCIb] (refs. 20,27) competitively inhibit ICN-Ndel1 binding[19], initiate dynein-dynactin binding (independent of a cargo adapter), and potentially relieves dynactin autoinhibition (the latter of which is due to

interactions between p150[CCI] and the pointed end complex)[25]. **5** Binding of a cargo adapter to the adapter-independent dynein-dynactin complex, which requires ICN-p150 binding, leads to assembly of the active dynein-dynactin-adapter (DDA) complex. Insets show AF2 models of human ICN with either Ndel1 or p150[CCI]. Note that data suggests that the ICN-p150 interaction involves both the SAH and H2 of ICN[27], but only the AF2 predicted H2-p150 interaction is shown here. **6** Coincident with microtubule binding, LIS1 dissociates[89], and the active DDA complex processively transports cargoes along microtubules.

for processive motility (Fig. 6, step 5; cargo not shown for clarity). Data indicate that LIS1 dissociates from the DDA complex as a consequence of dynein-microtubule binding immediately prior to commencement of motility[73,89–92] (Fig. 6, step 6). Although LIS1 has been observed comigrating with processively moving DDA complexes in vitro[52,66], it is unclear if these LIS1 molecules are bound to those dynein motors directly engaged with the microtubule, or possibly to a second dynein dimer scaffolded within the motile DDA complex that is not engaged directly in motility.

Very recent work using human proteins have demonstrated a similar capacity for Ndel1 to inhibit the assembly of motile DDA complexes[69], likely by competing with p150 for the ICN, as demonstrated here. However, in support of a positive role in dynein activity, another recent study found that Nde1-mediated recruitment of LIS1 to dynein indeed enhances DDA assembly[93]. In summary, our results provide new insights into previously unknown steps in the dynein activation pathway, provide a new model for the mysterious role of Ndel1 in the dynein pathway, and reveal a critical, evolutionarily conserved role for the ICN in the activation of dynein in vivo.

## Methods
### Media and strain construction
Yeast strains are derived from W303 or YEF473A[94] and are available upon request. We transformed yeast strains using the lithium acetate method[95]. Strains carrying mutations or tagged components were

constructed by PCR product-mediated transformation[96] or by mating followed by tetrad dissection. Proper tagging and mutagenesis were confirmed by PCR, and in some cases sequencing. Fluorescent tubulin-expressing yeast strains were generated using plasmids and strategies described previously[97] Strains overexpressing the yeast dynein complex (WT or ΔICN) were generated by transforming p8His-ZZ-SNAPf-Dynein or p8His-ZZ-HALO-Dynein (wild-type or mutants; see below and Supplementary Table 1) linearized by digestion with ApaI. Strains overexpressing yeast Nip100[CCI], Ndl1, or Pac1 were generated by transforming pRS306:GAL1p:EGFP-Nip100[CCI], pRS306:GAL1p:Ndl1-FLAG-SNAPf-TEV-ZZ-8xHis, or pRS306:GAL1p:8xHIS-ZZ-2xTEV-Pac1-FLAG-SNAPf (wild-type or mutants) linearized by digestion with ApaI. Integration was confirmed by diagnostic PCR. Yeast synthetic defined (SD) medium was obtained from Sunrise Science Products. (San Diego, CA).

### Plasmid construction
Rat IC2C (Uniprot ID:Q13409) WT and ΔICN with C-terminal SNAP-FLAG-strepII tag were cloned into pcDNA5-FRT-TO. Full-length human Ndel1 (Uniprot ID: Q9GZM8, or the coiled-coil domain, Ndel1[CC], residues 1-195) were cloned into pET28a with StrepII-sfGFP-PPS tags at its N-terminus, and FLAG tag at its C-terminus. Mutations in NudEL were generated by site-directed mutagenesis using hotstart Q5 DNA polymerase (New England Biolab). Human dynactin p150-CC1 (Uniprot ID: Q14203, residues 224-554) was cloned into pET28a with N-terminal

StrepII-SNAPf-PPS tags. The N-terminal coiled-coil domain of mouse BicD2 (residues 25–425) and the SNAPf-tagged and GST-dimerized human dynein motor domain were previously described[12]. SNAPf-tagged and 6xHis-tagged LIS1 expression constructs were previously described[66]. A plasmid encoding the entire yeast dynein complex[62] (p8His-ZZ-HALO-Dynein) was used to generate the ΔICN mutant (with N-terminal 43 residues of Pac11 deleted). An S6 tag (GDSLSWLLRLLN) was added to the C-terminus of Dyn2 in each of these plasmids (p8His-ZZ-HALO-Dynein and p8His-ZZ-HALO-Dynein^PacII-ΔICN). Full-length yeast Ndl1 with C-terminal FLAG-SNAPf-TEV-ZZ-8xHis tags was cloned into pRS306. The Ndl1 coiled-coil domain (residues 2–132) with N-terminal 6xHis-StrepII-sfGFP tags was cloned into pET28a. The Nip100 coiled-coil-1 (residues 97–377) were cloned into a pGEX-KG vector such that the Nip100^CC1 possessed an N-terminal SUMO-eGFP tag. Nip100^CC1 was also cloned into pRS306:GAL1p:EGFP. Full-length yeast Pac1 with N-terminal 8xHis-ZZ-2xTEV tags and C-terminal FLAG-SNAPf tags was described previously[89]. See Supplementary Table 1 for a list of oligo-nucleotides used throughout this study. All plasmids (and mutations as indicated throughout) were generated using Gibson assembly and the coding sequences were validated by DNA sequencing.

## Generation of IC2C Flp-In T-REx 293 cell lines and large scale expression

Flp-In™ T-REx™293 cells (Thermo Fisher, Cat#R78007) harboring IC2C wt or ΔICN with C-terminal SNAP-FLAG-strepII tagwhich were generated using the FLP/FRT system (Thermo Fisher). Briefly, pcDNA5-FRT-TO construct and pOG44, which expresses Flipase, were co-transfected using Lipofectamine 2000 (Thermo Fisher) into Flp-In™ T-REx™293 cells. After recovery from transfection, cells were grown in DMEM containing 10% FBS, 1% Penicillin-Streptomycin, and 100 μg/mL Hygromycin B to select cells in which pcDNA5 construct was inserted into FRT locus. The expression of tagged IC2C was verified by western blotting with antibodies against StrepII tag (Novus Biologicals). These cell lines were grown in culture vessels (Corning, Fisher Scientific) to 80% confluence, and then tagged IC2C expression was induced with 1 μg/ml doxycycline for two days. Cells were harvested in PBS by tapping the culture vessels, and spun down at 1200 rpm for 5 min in Sorvall Legend XTR. Cell pellets were washed with PBS, snap frozen in liquid nitrogen, and stored at −80 °C.

## Protein expression and purification

Porcine brain tubulin was isolated using the high-molarity PIPES procedure as described[98] and then labeled with biotin- or Dylight-405 NHS-ester (Invitrogen) as described (http://mitchison.hms.harvard.edu/files/mitchisonlab/files/labeling_tubulin_and_quantifying_labeling_stoichiometry.pdf). Microtubules were prepared by incubating tubulin with 1 mM GTP for 10 min at 37 °C, followed by diluting into 20 μM taxol and continuing incubation for an additional 20 min. Microtubules were pelleted through 25% sucrose cushion with 10 μM taxol at 80,000 rpm in a TLA-100 rotor, and the pellet was resuspended in BRB80 containing 10 μM taxol. Human dynein complexes were purified from Flp-In T-Rex cell pellet prepared as above. Cells were lysed in lysis buffer (50 mM Tris-HCl, pH 8.0, 150 mM KCH_3COO, 2 mM MgSO_4, 1 mM EGTA, 10% glycerol) supplemented with 0.2% Triton X-100, 0.1 mM ATP, 1 mM DTT, 1 mM PMSF, and DNaseI. Cell lysate was clarified at 147,000 × g for 15 min at 4 °C in Beckman MLA-50 rotor. The resulting supernatant was passed through HiPrep26/10 desalting column to remove free biotin, then loaded onto a column packed with strep-tactin XT 4Flow resin (Iba Lifesciences GmbH). The column was washed with 5 column volume of lysis buffer containing 250 mM NaCl followed by 10 column volume of lysis buffer. Proteins were eluted with 50 mM biotin, concentrated with an Amicon-Ultra MWCO 100 kDa filter, and then gel filtered on a Yarra 3 μm SEC-4000 LC column 300 ×21.2 mm (Phenomenex) equilibrated with GF150 buffer (25 mM HEPES-KOH, 150 mM KCl, 2 mM MgCl_2). Peak fractions

containing intact dynein were supplemented with 0.1 mM ATP, 1 mM DTT and 10% glycerol, and concentrated before being snap-frozen in liquid nitrogen, and stored at −80 °C. For labeling purified proteins with SNAP dyes, proteins were incubated with 2−5 molar excess of SNAP dye (SNAP- Alexa 647, SNAP-TMR star or SNAP-Surface 488) for 1 h on ice. The unbound dye was removed using HiTrap desalting columns (Thermo Fisher). The stoichiometry of labeling was assessed using a Nanodrop One (Thermo Fisher) and comparing the absorbance of total protein at 280 nm to the absorbance at the SNAP dye wavelength.

Porcine brain dynactin was purified according to[15]. Briefly, porcine brains were homogenized via blending in HB buffer (35 mM PIPES-KOH pH 7.2, 1 mM MgSO4, 0.2 mM EGTA, 0.1 mM EDTA, 1 mM DTT) supplemented with 2 mM PMSF and cleared in a Ti45 rotor (Beckman Coulter) at 235,000 rcf for 50 min. The resulting supernatant was the filtered through glass fiber and 0.45 μm syringe filters before loaded onto 300 mL SP-Sepharose Fast Flow (GE Healthcare) equilibrated in HB buffer supplemented with 0.1 mM ATP. Bound proteins were eluted with 0.5 M KCl. Dynactin containing fractions were loaded onto a MonoQ HR 10/100 column (GE Healthcare) and bound proteins were eluted in three phase linear gradient 50−150 mM, 150−350 mM, and then 350−1 M KCl gradient. Dynactin containing fractions were concentrated and loaded on a Yarra 3 μm SEC-4000 LC column 300 × 21.2 mm (Phenomenex) equilibrated with GF150 buffer. Peak fractions containing dynactin were supplemented with 0.1 mM ATP, 1 mM DTT and 10% glycerol, and concentrated before being snap-frozen in liquid nitrogen, and stored at −80 °C. Purified BicD2 and Hook3 1-552 were used to isolate DDA complexes from rat brain cytosol as previously described[12]. Briefly, 300 nM of adapter protein was added to the brain lysate and incubated at 4 °C for 1 h with gentle agitation. Resulting DDA complex was pulled down with strep-tactin beads (IBA Lifesciences GmbH). DDA complexes were labeled with 5 μM SNAP-TMR dye during the isolation procedure, snap-frozen in small aliquots, and stored at −80 °C. Bacterial expression constructs for BicD2, Hook3 1-552, NudEL-FL, NudEL^CC, and p150^CC1 were transformed into BL21-CodonPlus (DE3)-RIPL (Agilent) and the bacteria were grown in LB medium at 37 °C until an OD_600 of 0.6. The protein expression was induced with 0.2 mM isopropyl-β-D-thiogalactoside overnight at 18 °C, except the p150^CC1 construct which was induced for 2 hours at 37 °C. Cells were harvested and resuspended in lysis buffer supplemented with 1 mM DTT, 1 mM PMSF, DNaseI. Cells were lysed by passing through an Emulsiflex C3 high-pressure homogenizer (Avestin). Then the lysates were centrifuged at 15,000 × g for 20 min at 4 °C and the supernatant was passed over a column packed with Strep-tactin XT 4Flow resin (Iba Lifesciences GmbH). The column was washed with lysis buffer, and bound proteins were eluted in lysis buffer containing 50 mM biotin (Chem-Impex International). For StrepII-sfGFP cleaved NudEL-FL and NudEL^CC used in Fig. 5e, f, the proteins were cleaved while bound to the columns with Prescission protease, and cleaved NudEL-FL and NudEL^CC were collected. Eluted proteins were directly loaded onto HiTrap Q HP column (GE Healthcare) and eluted with 0−0.6 M NaCl gradient. Peak fractions were concentrated on Amicon-Ultra filters and passed through an EnRich650 (Bio-Rad) or Superpose 6 10/300 GL (GE Healthcare) size exclusion column, in lysis buffer. Peak fractions were collected, concentrated again, and frozen after supplemented with 1 mM DTT and 10% glycerol. StrepII-SNAPf-LIS1 was expressed in Sf9 cells and purified as described above for the bactrially expressed proteins. 6xhis-LIS1 was also expressed in Sf9 cells and purified first with Ni-NTA resin according to the manufacturer's condition, followed by HiTrapQ column and EnRich650 column as described above.

Purification of the yeast dynein complex (6xHis-ZZ-TEV-HALO-or SNAPf-Dynein, with all genes, including Dyn2, Dyn3 and Pac11 under the control of the GAL1p promoter) was performed as previously described with minor modifications[62]. In brief, yeast cultures were grown in YPA medium supplemented with 2% galactose for no more

than 3 h, collected, washed with cold water and then resuspended in a small volume of water. The resuspended cell pellet was drop-frozen into liquid nitrogen and then lysed in a coffee grinder. After lysis, 0.25 volumes of 4X dynein lysis buffer (1X buffer: 30 mM HEPES-KOH pH 7.2, 50 mM potassium acetate, 2 mM magnesium acetate, 0.2 mM EGTA) supplemented with 1 mM dithiothreitol (DTT), 0.1 mM Mg-ATP and 0.5 mM Pefabloc SC or protease inhibitor tablets (Pierce) (concentrations for the 1X buffer) was added, and the lysate was clarified by centrifugation at 310,000 × $g$ for 1 h. The supernatant was then incubated with IgG sepharose 6 fast flow resin (GE) for 1-2 h at 4 °C, which was subsequently washed three times in 5 ml lysis buffer, and twice in 5 ml TEV buffer (50 mM Tris-HCl pH 8.0, 150 mM potassium acetate, 2 mM magnesium acetate, 1 mM EGTA and 10% glycerol) supplemented with 0.005% Triton X-100, 1 mM DTT, 0.1 mM Mg-ATP and 0.5 mM Pefabloc SC. To fluorescently label the dyneins, the bead-bound protein was incubated with 5–10 μM JF503-HaloTag, JFX549-HaloTag, or JFX646-HaloTag ligand (Janelia Research Campus), as appropriate, for 10–20 min at room temperature. The resin was then washed four more times in TEV buffer supplemented with 1 mM DTT, 0.005% Triton X-100 and 0.1 mM Mg-ATP, and then incubated with TEV protease for 1–1.5 h at 16 °C. After digestion with TEV, the beads were pelleted, and the resulting supernatant was collected, aliquoted, flash-frozen in liquid nitrogen, and stored at −80 °C. Protein preparations used for negative stain EM imaging (6xHis-ZZ-TEV-SNAPf-Dynein) were tandem-affinity purified. To do so, subsequent to lysis, 0.25 volumes of 4X NiNTA dynein lysis buffer (1X buffer: 30 mM HEPES pH 7.2, 200 mM potassium acetate, 2 mM magnesium acetate and 10% glycerol) supplemented with 1 mM beta-mercaptoethanol, 0.1 mM Mg-ATP and 0.5 mM Pefabloc SC (concentrations for the 1X buffer) was added, and the lysate was clarified as described above. The supernatant was then bound to NiNTA agarose for 1 h at 4 °C, which was subsequently washed three times in 5 ml NiNTA lysis buffer. The protein was eluted in NiNTA lysis buffer supplemented with 250 mM imidazole by incubation for 10 min on ice. The eluate was then diluted with an equal volume of dynein lysis buffer, which was then incubated with IgG Sepharose 6 fast flow resin for 1 h at 4 °C. The beads were washed and the protein was eluted as described above (with TEV protease). Eluted protein was either applied to a size-exclusion resin (Superose 6; GE) or snap-frozen. The gel filtration resin was equilibrated in GF150 buffer (25 mM HEPES-KOH pH 7.4, 150 mM KCl, 1 mM MgCl₂, 5 mM DTT and 0.1 mM Mg-ATP) using an AKTA Pure system. Peak fractions (determined by absorbance at 260 nm and SDS–PAGE) were pooled, concentrated, aliquoted, flash-frozen and then stored at −80 °C.

Purification of Ndl1-SNAPf-TEV-ZZ was performed using the same procedure as that for ZZ-TEV-Pac1-SNAPf as previously described[62], with the addition of a gel filtration step to remove residual, unbound fluorescent dye. Specifically, proteins were fluorescently labeled by incubating the bead-bound protein with either 10 μM JFX554-SNAP or JFX650-SNAP ligand (Janelia Research Campus), as appropriate, for 1 hour at 4 °C. The resin was then washed four times in TEV buffer supplemented with 0.005% Triton X-100, 1 mM DTT and 0.5 mM Pefabloc SC, then incubated in TEV buffer supplemented with TEV protease overnight at 4 °C. Following TEV digest, the supernatant was collected using a spin filtration device, and applied to a size exclusion chromatography resin (Superose 6; GE) that was equilibrated in TEV buffer supplemented with 1 mM DTT using an AKTA Pure. Peak fractions (determined by absorbance at 260 nm and SDS-PAGE) were pooled, concentrated, aliquoted, flash-frozen and then stored at −80 °C.

To purify StrepII-sfGFP-Ndl1^CC (residues 2–132), cultures of BL21-CodonPlus (DE3)-RIPL containing the plasmid were grown at 37 °C until an OD₆₀₀ of ~0.8 at which point protein expression was induced with 1 mM IPTG, and the temperature was shifted to 16 °C. After an overnight incubation, cells were harvested, washed with water, and

frozen. Cell pellets were resuspended in buffer W (100 mM Tris-HCl, pH 8.0, 150 mM NaCl, 1 mM EDTA) by passage through a microfluidizer three times at 18,000 psi. The cell lysate was clarified by centrifugation at 23,000 × $g$ for 20 min, and the clarified lysate was then loaded onto a gravity flow column packed with Strep-tactin XT agarose beads (Iba Lifesciences). After binding, the column was washed extensively, and the bound protein was eluted in buffer W supplemented with 50 mM biotin. After elution, peak fractions were concentrated, and gel filtered using a Superdex 200 16/60 gel filtration column (GE Healthcare). The peak fractions were concentrated, flash frozen, and then stored at −80 °C.

To purify GST-eGFP-Nip100^CC1 (residues 97–377), cultures of BL21-CodonPlus (DE3)-RIPL containing the plasmid were grown and harvested as described above. Cell pellets were resuspended in buffer W by passage through a microfluidizer three times at 18,000 psi. The cell lysate was clarified by centrifugation at 310,000 × $g$ for 1 h. The supernatant was incubated with Glutathione agarose 4 resin for 1–2 h at 4 °C, which was subsequently washed three times in 5 ml buffer W. The protein-containing resin was then incubated with Ulp1 protease for 1–1.5 h at 16 °C. Following Ulp1 digest, the supernatant was collected using a spin filtration device, and gel filtered using a Superdex 200 16/60 gel filtration column (GE Healthcare). The peak fractions were concentrated, flash frozen, and then stored at −80 °C.

## Single and ensemble molecule motility assays

The single-molecule motility assays with yeast proteins were performed using previously reported protocols[99,100] with minor modifications. Briefly, coverslips were cleaned with acetone and potassium hydroxide, followed by oxygen plasma treatment (for 10 min). The coverslips were silanized with 3-aminopropyl trimethoxysilane (APTES), and then coated with a biotinylated polyethylene glycol (PEG) solution (a mixture of methoxyPEG-succinimidyl valerate and biotin-PEG-succinimidyl valerate dissolved in a freshly prepared solution of 0.1 M sodium bicarbonate). The coverslips were adhered to glass slides using double-sided adhesive tape, thereby producing narrow chambers (~4–7 μl in volume). The flow chambers were incubated with streptavidin (0.1 mg/ml), and then blocked with 1% Pluronic F-127. Taxol-stabilized, biotinylated microtubules assembled from unlabeled and biotin-labeled porcine tubulin (4:1 ratio) were introduced into the chamber. Following a 5–10 min incubation, the chamber was washed with dynein lysis buffer supplemented with 20 μM taxol, after which dynein diluted in an oxygen-scavenging motility buffer (30 mM HEPES, pH 7.2, 50 mM potassium acetate, 2 mM magnesium acetate, 1 mM EGTA, 10% glycerol, 50 nM protocatechuate 3,4-dioxygenase, 2.5 mM protocatechuic acid, 1 mM Trolox, 1 mM cyclooctatetraene, 1 mM 4-nitrobenzoyl alcohol) supplemented with 1 mM DTT, 20 μM taxol, and 1 mM Mg-ATP was added. TIRFM images were immediately collected using a 1.49 NA 100X TIRF objective on a Nikon Ti-E inverted microscope equipped with a Ti-S-E motorized stage, piezo Z-control (Physik Instrumente), and an iXon X3 DU897 cooled EM-CCD camera (Andor). 488 nm, 532 nm, and 640 nm lasers were used along with a multi-pass quad filter cube set (C-TIRF for 405/488/561/638 nm; Chroma) and emission filters mounted in a filter wheel (525/50 nm, 600/50 nm and 700/75 nm; Chroma). We acquired images at 1 second intervals for 8 min. Velocity and run length values were determined from kymographs generated using the MultipleKymograph plugin for ImageJ (http://www.embl.de/eamnet/html/body_kymograph.html). Mean run length values for individual runs were determined by fitting the cumulative distribution functions to a one-phase decay in Graph-Pad Prism, as previously described[62,101].

Single molecule assays with mammalian proteins were performed in chambers with surface adhered porcine microtubules essentially as previously described[102,103]. The buffer used for these assays was SRP90 (90 mM HEPES-KOH pH7.4, 50 mM K- acetate, 2 mM Mg-acetate, 1 mM EGTA, 10% glycerol supplemented with 0.5 % Pluronic F-127, 0.1 mg/ml

biotin-BSA, 0.2 mg/ml BSA, 0.2 mg/ml κ-casein, 10 μM Taxol, 2 mM Trolox, 2 mM protocatechuic acid, 50 nM protocatechuate-3,4-dioxygenase, 2 mM Mg-ATP).

For microtubule gliding assays, 0.5 μM human dynein was flowed into empty chambers. After incubating the chamber for 5 min, unbound proteins were washed away with SRP90 assay buffer. Dylight-405-labeled microtubules diluted in assay buffer were flowed in, and images were acquired every 5 s. Kymographs of microtubules were prepared using ImageJ software, and velocities were determined based on translocation over time. For DDH single molecule motility assays, 30 nM DDH and indicated concentrations of either p150$^{CCl}$ or Ndel1 in the SRP90 assay buffer were flowed into the glass chamber. Images were acquired every 0.5 sec for 2 min. All images were acquired using a Nikon TE microscope (1.493NA, 100X objective) equipped with a TIRF illuminator and Andor iXon CCD EM camera.

## Microtubule recruitment assays

For yeast proteins, flow chambers constructed using slides and plasma cleaned and silanized coverslips attached with double-sided adhesive tape were coated with anti-tubulin antibody (8 mg/ml, YL1/2; Accurate Chemical & Scientific Corporation) then blocked with 1% Pluronic F-127 (Fisher Scientific). Taxol-stabilized microtubules assembled from unlabeled porcine tubulin (Cytoskeleton) were introduced into the chamber. After microtubules were adhered to the cover glass, the chambers were washed with TEV buffer supplemented with 1 mM DTT and 20 μM taxol. Following this, mixtures of purified proteins (e.g., dynein ±Ndl1; as described throughout the text and in figure legends) were flowed into the chambers, after which the chambers were incubated for 5–10 min and then imaged. Quantitation was performed using ImageJ/FIJI (National Institutes of Health). Fluorescence intensities in each channel were measured along microtubules ("signal"), and adjacent to microtubules ("background"). Mean corrected pixel intensity was determined by subtracting background from signal.

Assays with mammalian proteins were carried out as follows: a mixture of unlabeled tubulin, biotin-tubulin, and DyLight405-labeled microtubules were prepared as described. Microtubules were pelleted over a 25% sucrose cushion in BRB80 buffer to remove unpolymerized tubulin. TIRF chambers were assembled from acid-washed coverslips and double-sided sticky tape. Chambers were first incubated with 0.5 mg/ml PLL-PEG-Biotin (Surface Solutions) for 10 min, followed by 0.5 mg/ml streptavidin for 5 min. MTs diluted into taxol containing BRB80 buffer were then incubated in the chamber and allowed to adhere to the streptavidin-coated surface. Unbound MTs were washed away with SRP90 assay buffer. Proteins (e.g., dynein, LIS1, Ndel1) were diluted in assay buffer at concentrations indicated in Figure legends supplemented with 2 mM AMP-PNP (Roche). Prior to image acquisition, chambers were incubated 10 min to allow proteins to reach steady-state.

## Live cell imaging experiments

For the single timepoint spindle position assay, the percentage of cells with a misoriented anaphase spindle was determined after growth overnight (12–16 h) at a low temperature (16 °C), as previously described[54,58,104]. A single z-stack of wide-field fluorescence images was acquired for mTurquoise2- or mRuby2-Tub1, as appropriate. For the spindle oscillation assay[59], cells were arrested with 200 mM hydroxyurea (HU) for 2.5 h, and then mounted on agarose pads containing HU for fluorescence microscopy. GFP-labeled microtubules (GFP-Tub1) were imaged every 10 s for 10 min. To image dynein localization, cells were grown in synthetic defined (SD) media supplemented with either glucose, raffinose, or raffinose plus galactose (for GAL1p experiments; induced for 3.5 h), and mounted on agarose pads. Images were collected on a Nikon Ti-E microscope equipped with a 1.49 NA

100X TIRF objective, a Ti-S-E motorized stage, piezo Z-control (Physik Instrumente), an iXon DU897 cooled EM-CCD camera (Andor) with an emission filter wheel (ET480/40M for mTurquoise2, ET525/50M for GFP, ET520/40M for YFP, and ET632/60M for mRuby2; Chroma). The microscope was controlled with NIS Elements software (Nikon). Image analysis was performed using ImageJ/FIJI software (National Institutes of Health). Plus end, cortical, and SPB foci were identified in two-color movies and scored accordingly. Specifically, plus end molecules were recognized as those foci that localized to the distal tips of dynamic microtubules (identified via mTurquoise2- or mRuby2-Tub1 imaging), whereas spindle pole body (SPB)-associated molecules were recognized as those foci that localized to one of the spindle poles. Cortical molecules were identified as those foci not associated with an astral microtubule plus end that remained stationary at the cell cortex for at least three frames.

## Mitotic index measurement

T-Rex Flp-In 293 cells were grown on coverslips, and DIC (WT or ΔICN) expression was induced with 1 μg/ml doxycycline for 2 days. Cells (~60% confluent) were fixed with 4% paraformaldehyde for 15 min, rinsed with PBS, and permeabilized subsequently with 0.1% Triton-X100/PBS. Cells were stained with anti-tubulin antibody DM1A (Cell Signaling, 1:2000 dilution), anti-SNAP tag (Invitrogen, 1:2000 dilution), and DAPI. Cell images were collected from 5 randomly selected fields and mitotic cells were scored.

## DDH reconstitution from purified proteins

Purified dynein, pig brain dynactin, Hook3 (1-552), and LIS1 were diluted into assembly buffer (90 mM HEPES-KOH pH 7.6, 50 mM KCH3COO, 2 mM Mg (CH3COO)$_2$, 1 mM EGTA, 10% glycerol, 0.1% Triton X-100) at final concentrations of 30 nM, 30 nM, 200 nM, and 0.8 μM, respectively. The mixture was incubated for 1 h at 4 °C with gentle agitation. After the incubation, 20 μl anti-FLAG M2 agarose beads (Sigma) were added in order to pull down DDH and/or remaining dynein through FLAG-tag on IC2C subunit of dynein. The mixture was incubated for additional 1 h at 4 °C with gentle agitation, and then beads were washed three times with assembly buffer. Protein bound to the beads were analyzed by SDS-PAGE and western blotting. Antibodies used for westernblotting are: anti-p150 (BD Biosciences Cat#612709), anti-dynein intermediate chain (clone 74.1, Sigma-Aldrich, Cat# MAB1617), anti-LIS1 (Sigma-Aldrich, Cat# L-7391), and anti-StepII-tag (Novus Biochemicals Cat#NBP2-41076) for Hook3. All antibodies were diluted 1:2000 for westernblotting.

## Mass photometery

To prepare chambers, high precision microscope coverslips (No. 1.5, 24 × 50, cat# 0107222, Marienfeld, Lauda-K önigshofen, Germany) were cleaned by sequential sonication in ultrapure H$_2$O (10 min), isopropanol (10 min), then ultrapure H$_2$O, and dried with filtered air. Culturewell$^{TM}$ gaskets (3 mm diam × 1 mm depth, cat#103250, Grace Bio-Labs, Bend, OR) were cut and rinsed with isopropanol and ultrapure H$_2$O, then dried with filtered air and placed onto the freshly cleaned cover glasses. To focus, 15 μl buffer (90 mM HEPES-KOH pH7.4, 50 mM K- acetate, 2 mM Mg-acetate, 1 mM EGTA, 10% glycerol supplemented) was first applied to the well, and the focal position was identified and secured in place with an autofocus system based on total internal reflection for the entire measurement period. Proteins were preincubated at 0.5–1 μM for 5 min at room temperature, followed by immediate dilution in the same buffer. Next, 5 μl of diluted protein was added into the measurement well (5–20 nM final concentration). All data were acquired using an OneMP mass photometer (Refeyn Ltd, Oxford, UK) at the rate of 1 kHz for 60 s by AcquireMP (Refeyn Ltd). Each sample was measured at least three times independently. Calibration standard was generated using BSA

(Sigma), Alcohol dehydrogenase (A7011, Sigma) and beta-amylase (A8781, Sigma). Data analysis was performed using DiscoverMP (Refeyn Ltd).

Mass photometry data with yeast proteins (Pac1 and Ndl1$^{CC}$) were performed similarly, but with a TwoMP mass photometer (Refeyn Ltd, Oxford, UK). All proteins were initially diluted to 100–200 nM in assay buffer (50 mM Tris, pH 8.0, 150 mM potassium acetate, 2 mM magnesium acetate, 1 mM EGTA, 1 mM DTT; see Figure legends). 2 µl of each was then mixed 1:1 (to 50–100 nM of each), incubated for 1–2 min, and then diluted 1:5 on the stage (2.5 µl of mixed protein +10 µl assay buffer) to 10–20 nM (final concentration) immediately prior to image acquisition. For measurements of individual proteins, 100–200 nM of each were diluted 1:10 on the stage (1.5 µl of protein + 13.5 µl assay buffer; 10–20 nM final concentration). 1 min movies were acquired using AcquireMP, and all images were processed and analyzed using DiscoverMP

### Negative stain EM and image analysis
TEM grids were prepared by applying fresh dynein samples (WT or ΔICN, after tandem affinity purification; see above) to glow-discharged carbon-coated 200 mesh copper grids. After incubation for ~1 min, 2% uranyl acetate was added. Micrographs were collected with a FEI Tecnai F20 200 kV TEM equipped with a Gatan US4000 CCD (model 984), at a nominal magnification of X90,000 with a digital pixel size of 6.19 Å. All image analysis was performed using Relion v.3.0 on the University of Colorado Boulder High Performance Computer Cluster, Summit. Particles were manually picked from ~20 micrographs (~200 particles), which were used to generate a low-resolution 2D class average. Using these 2D averages as a starting point, we then used an iterative process to autopick particles that were used to generate the final 2D averages

### AlphaFold predictions
Although a crystal structure of the coiled-coil domain of Ndel1 revealed a tetramer[65], we chose to model these molecules as dimers in light of our mass photometry data. For our models with Pac1 or LIS1, we used a stoichiometry of 1 Ndel1/Ndl1 dimer to 1 LIS1/Pac1 dimer, which was also guided by our mass photometry data. Finally, for our Ndel1/Ndl1·ICN complexes – which were modeled without LIS1/Pac1 – we chose to use a stoichiometry of 1 Ndel1/Ndl1 to 2 ICNs given the likely presence of two ICN-binding sites within a Ndel1/Ndl1 dimer. Models were generated using ColabFold running on Google Colaboratory[64], and images were manipulated and prepared using ChimeraX[105].

### Statistical analyses
All data were collected from at least two independent replicates (independent protein preparations or cell cultures for in vitro and in vivo experiments, respectively). The values from each independent replicate – which are indicated on each plot – showed similar results. For all datasets, *P* values were calculated from Z scores (when comparing proportions) as previously described[60], or by performing unpaired two-tailed Welch's *t*-test, or the Mann–Whitney test, the latter two of which were performed using GraphPad Prism. These latter tests were selected as follows: the unpaired two-tailed Welch's *t*-test was used when the datasets in question were both determined to be normal (by the D'Agostino and Pearson test for normality; *P* > 0.05); in the case where only one (or neither) of the datasets were determined to be normal (*P* < 0.05), the Mann–Whitney test was used. Z scores were calculated using the following formula:

$$Z = \frac{(\hat{p}_1 - \hat{p}_2)}{\hat{p}(1 - \hat{p})\left(\frac{1}{n_1} + \frac{1}{n_2}\right)}$$

where:

$$\hat{p} = \frac{y_1 + y_2}{n_1 + n_2}$$

Z scores were converted to two-tailed *P* values using an online calculator.

### Reporting summary
Further information on research design is available in the Nature Portfolio Reporting Summary linked to this article.

## Data availability
Due to the large volume of raw data generated throughout this study, they are available from the authors only upon request. Source analysis data for all figures are included in the accompanying Source Data file. Source data are provided with this paper.

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

## Acknowledgements

We are grateful to members of the McKenney, Ori-McKenney, Markus, DeLuca, DeSantis, and Yildiz labs for valuable discussions and sharing of data. Electron microscopy was done at the University of Colorado, Boulder EM Services Core Facility in the MCDB Department, with the technical assistance of facility staff, or in the Biological Electron Microscopy Facility at UC Davis. This work utilized the RMACC Summit supercomputer, which is supported by the National Science Foundation (awards ACI-1532235 and ACI-1532236), the University of Colorado Boulder and Colorado State University. The RMACC Summit supercomputer is a joint effort of the University of Colorado Boulder and Colorado State University. This work was funded by the NIH/NIGMS (2R35GM124889 and 4R00NS089428 to R.J.M, and R35GM139483 to S.M.M.).

## Author contributions

R.J.M and S.M.M. designed the study, acquired funding, generated figures, and wrote and edited the manuscript. K.O. and B.I. generated reagents, performed experiments, analyzed the in-cell, in vivo, and in vitro data, helped prepare figures, and edited the manuscript. L.L. performed and analyzed the spindle oscillation experiments. W.L. and P.G. generated reagents for work with metazoan proteins and performed initial TIRF experiments. All authors edited the paper.

## Competing interests

The authors declare no competing interests.
