## [Peer Review File · Nature Communications]

REVIEWER COMMENTS

Reviewer #1 (Remarks to the Author):

In this study, Okada et al. investigate the role of the dynein-associated protein Ndel1 in the assembly of dynein-dynactin-adaptor (DDA) motile complexes. Working with both human and yeast dynein, the authors look into the effects of mutations and truncations of the Dynein Intermediate Chain N-terminus (ICN), a binding partner for Ndel1, and find that it is required for dynein function *in vivo*, but not for activity in TIRF assays. Using *in vitro* methods, the authors show that the ICN is involved in the assembly of DDA complexes, possibly through a known direct interaction with the dynactin subunit p150. After showing that ICN-bound Ndel1 is necessary for the recruitment of the dynein activator Lis1 to dynein, the authors use AlphaFold modelling to predict the interaction sites between dynein, Ndel1, and Lis1, and validate the model with point mutants. Finally, the authors dissect how Ndel1 competitively interferes with the interaction between ICN and dynactin p150. Overall, Okada et al. present a well-executed study that significantly advances our understanding of the molecular mechanisms governing dynein-dynactin assembly and function, which will be of significant interest to the broad audience of molecular motors.

The mass photometry data presented in this paper serves to validate the structure of the interaction between Ndel1 and Lis1, and the role of Ndel1 in inhibiting the interaction of Lis1 with the dynein motor domain. The experiments are competently performed, and showcase a novel application of this technique.

Minor points:

- Figure 4E: the relative concentrations used for the MP experiment are not clear. Clearly specifying in the legend whether a 1:1:1 ratio was used of all components, or whether there was an excess of Ndel1, could assist the reader in understanding the results.
- Figure 5G: including controls of these motility assays in which the complexes were assembled in presence of p150CC1 or Ndel1 might further corroborate the observations, and provide an interesting comparison in which co-migration might be increased.

Reviewer #2 (Remarks to the Author):

This study by Okada and coworkers explores the confusing ability of the dynein intermediate chains (ICs) to bind both dynactin (via P150Glued) and Nde1 through the same domain, both of which are required for dynein motility. Structural studies have failed to resolve the relevant portions of these proteins, preventing a compelling molecular analysis of these interactions. The authors generate a series of mutant dynein constructs that are designed to tease apart distinct subcomplexes which are then tested in *in vitro* motility assays. These studies suggest that assembly of a functional dynein-dynactin-adaptor (DDA) complex requires the sequential recruitment of Nde1 to dynein via the dynein ICs which brings LIS-1 enabling the stabilization of the dynein motor domains into the open or active conformation. This step then allows Nde1 to release from the dynein ICs, which then allows binding and release of autoinhibition of dynactin. The addition of a cargo adaptor (ie. Hook3) then associates this complex with a cargo. The subsequent release of LIS-1 allows motility. This multi-step pathway for dynein activation is conserved from yeast to mammals, suggesting an ancient mechanism for dynein activation.

Overall, this Reviewer finds the model compelling and consistent with previous work in the field. The elucidation of a sequential recruitment/activation pathway for this multi-component system is a significant advance for this area of research and potentially explains previous contradictions that have stalled progress. This is an impressive advance for such a complicated pathway. The major concern I have for this model is that it requires multiple dramatic transitions in protein-protein interactions and the basis of these transitions is not investigated. Experiments that interfere with these transitions would strengthen the model considerably.

Concerns/Suggestions:

- 1) Although this Reviewer understands why the authors intermingled the experiments using mammalian dynein and yeast dynein, I found the jumping back and forth between model systems very distracting. This reviewer views the role of anchored dynein (ie. yeast) as being a little different in mechanism and protein composition from motile dynein (ie. membrane transport). Needing to think

about this for every experiment was challenging. In some ways the yeast dynein experiments and mammalian dynein experiments belong in separate papers. However, if the authors want to maintain the current style, I might recommend adding a Table (multi-column) in which they provide: 1) the mutant used, 2) the phenotype expected, and 3) some scoring scheme that indicates outcome. Lining up comparable mammalian and yeast constructs in the same rows would make interpretation easier.

2) A number of important molecular transitions are suggested in the model proposed by the authors. These include Nde1 releasing LIS-1 for binding to the dynein motor domain, Nde1 releasing from the dynein IC N-terminal domain, binding of dynactin (P150Glued) to the same N-terminal domain of the dynein ICs, recruitment of the cargo adaptor (ie. Hook3) and release of LIS-1 by the dynein motor domain for motility. This sequential activation model is very interesting and compelling as a mechanism. However, the molecular basis of these transitions is not really explored to any extent. Simple diffusion-based equilibria are possible, but not very tangible. Given that the candidate proteins at the core of this model are all known to be phosphorylation substrates, it is surprising that the role of phosphorylation wasn't incorporated into the model. The molecular consequences of some of these phosphorylation events are known and phosphorylation sites have been mapped and mutated. The addition of mutants that affect phosphorylation status would be a powerful addition to the study. Especially if these mutants block transitions.

Reviewer #3 (Remarks to the Author):

Okada and Iyer et al. set out to understand the role that dynein intermediate chain plays in dynein activation. The authors explore this question with human and yeast systems using a combination of biochemistry, cell biology, and microscopy. The authors find that dynein's ICD plays an important role in promoting the formation of dynein-dynactin-adaptor complexes. This finding is important because the role of IC-p150 binding in dynein activation is ill-defined since the regions of these proteins are not resolved in any published structures. The authors also find that Nde1 antagonizes the formation of DDA complexes by binding competitively with p150 subunit of dynactin. Finally, the authors find that though Nde1 binds competitively with dynein for Lis1 binding. This is a surprising finding that is in opposition to a well-established model of Lis1-Nde1 function. These are important findings that address several important and outstanding questions. All the work was performed to a high standard and supports the claims that the authors make. I recommend this for publication.

Recommendations:

1. It is easy to get lost in the descriptions of interactions between Nde1, dynein's IC, and dynactin's p150. Domain schematics for Nde1 and p150 would be helpful in Figure 1 (as the authors have done for IC).
2. I couldn't find the experimental methods for the experiment in 2D and struggled to follow the experimental design. Can the authors add the methods for this assay? What data is plotted in the graph in 2D? Is that with or without added Hook3? Is it surprising that inclusion of Hook3 did not increase the apparent dynein-dynactin association with WT dynein?
3. The authors should show the Alphafold-generated ICD domain-docked on Nde1 in the main text (some version of Figure S3A and B).
4. Should cite Garrot and Gillies, JBC 2023 article to support the observations that Nde1 disfavors DDA assembly and that Lis1 doesn't bind Nde1 and dynein simultaneously.

REVIEWER COMMENTS

We thank the reviewers for the time and effort they put into reading our manuscript, and for their constructive comments and insights about our work. We are appreciative that they view the work as rigorous and impactful for the molecular motors and cell biology fields.

Reviewer #1 (Remarks to the Author):

In this study, Okada et al. investigate the role of the dynein-associated protein Ndel1 in the assembly of dynein-dynactin-adaptor (DDA) motile complexes. Working with both human and yeast dynein, the authors look into the effects of mutations and truncations of the Dynein Intermediate Chain N-terminus (ICN), a binding partner for Ndel1, and find that it is required for dynein function in vivo, but not for activity in TIRF assays. Using in vitro methods, the authors show that the ICN is involved in the assembly of DDA complexes, possibly through a known direct interaction with the dynactin subunit p150. After showing that ICN-bound Ndel1 is necessary for the recruitment of the dynein activator Lis1 to dynein, the authors use AlphaFold modelling to predict the interaction sites between dynein, Ndel1, and Lis1, and validate the model with point mutants. Finally, the authors dissect how Ndel1 competitively interferes with the interaction between ICN and dynactin p150. Overall, Okada et al. present a well-executed study that significantly advances our understanding of the molecular mechanisms governing dynein-dynactin assembly and function, which will be of significant interest to the broad audience of molecular motors.

The mass photometry data presented in this paper serves to validate the structure of the interaction between Ndel1 and Lis1, and the role of Ndel1 in inhibiting the interaction of Lis1 with the dynein motor domain. The experiments are competently performed, and showcase a novel application of this technique.

Minor points:

- Figure 4E: the relative concentrations used for the MP experiment are not clear. Clearly specifying in the legend whether a 1:1:1 ratio was used of all components, or whether there was an excess of Ndel1, could assist the reader in understanding the results.

We apologize for the omission of this information and have now included the protein concentrations used in our mass photometry experiments (see new Figure 4 and S3 legends and updated methods).

- Figure 5G: including controls of these motility assays in which the complexes were assembled in presence of p150^{CC1} or Ndel1 might further corroborate the observations, and provide an interesting comparison in which co-migration might be increased.

In this experiment, we added exogenous p150^{CC1} or Ndel1 to purified DDH complexes and imaged the extent of their interactions using TIRF microscopy. The reason we added p150^{CC1} or Ndel1 after DDH complex assembly is that we found their addition during assembly strongly disfavors DDH formation (Fig. 5D-E), and thus would be expected to result in a large decrease in observable motion of intact DDH complexes. We would like to note that the data in Figure 5D-E showing Ndel1 inhibition of DDA assembly is consistent with recent data from Garrott and Gillies et al. (JBC, 2023), which was published as our manuscript was under review. Additionally, an experiment similar to the one suggested by the reviewer can also be found in Figure 4 of Zhao et al. BioRxiv, 2023

(<https://www.biorxiv.org/content/10.1101/2023.05.26.542537v1>), in which the authors (1) observe disruption of DDA assembly when exogenous Nde1 was added during complex assembly (similar to our findings shown in Figure 5D-E), and (2) do not observe co-migration of Nde1 with DDA complexes, also similar to our current data. This paper was co-submitted with ours to this journal.

To directly address the reviewer's question, we have performed the experiment again, but this time incubating p150^{CC1} or Nde1 with Hook3 during assembly of the DDH complex. We then examined the resulting complexes after isolation by Hook3 pulldown and subsequent TIRF microscopy. As seen below in Reviewer Figure 1, we observe few, if any, motile DDH complexes when either p150^{CC1} or Nde1 are present during assembly. These results mirror the biochemical results we present in Figure 5D-E, and support our model whereby the ICN is likely unavailable to bind to these proteins once DDH complex assembly is complete. Since this result is consistent with data presented in Figure 5D-E, we opted not to include it in our revised manuscript, but can place it in the supplement if the reviewer thinks it would be useful.

Reviewer #2 (Remarks to the Author):

This study by Okada and coworkers explores the confusing ability of the dynein intermediate chains (ICs) to bind both dynactin (via P150Glued) and Nde1 through the same domain, both of which are required for dynein motility. Structural studies have failed to resolve the relevant portions of these proteins, preventing a compelling molecular analysis of these interactions. The authors generate a series of mutant dynein constructs that are designed to tease apart distinct subcomplexes which are then tested in in vitro motility assays. These studies suggest that assembly of a functional dynein-dynactin-adaptor (DDA) complex requires the sequential

recruitment of Nde1 to dynein via the dynein ICs which brings LIS-1 enabling the stabilization of the dynein motor domains into the open or active conformation. This step then allows Nde1 to release from the dynein ICs, which then allows binding and release of autoinhibition of dynactin. The addition of a cargo adaptor (ie. Hook3) then associates this complex with a cargo. The subsequent release of LIS-1 allows motility. This multi-step pathway for dynein activation is conserved from yeast to mammals, suggesting an ancient mechanism for dynein activation. Overall, this Reviewer finds the model compelling and consistent with previous work in the field. The elucidation of a sequential recruitment/activation pathway for this multi-component system is a significant advance for this area of research and potentially explains previous contradictions that have stalled progress. This is an impressive advance for such a complicated pathway. The major concern I have for this model is that it requires multiple dramatic transitions in protein-protein interactions and the basis of these transitions is not investigated. Experiments that interfere with these transitions would strengthen the model considerably.

We thank the reviewer for their thorough reading of our paper and their helpful suggestions. We agree that there remain unanswered questions with regards to our proposed model, in particular those that revolve around how protein-protein interactions must change throughout the dynein activation process. These questions are very interesting, but also difficult to answer, and will require significant additional efforts to unravel the complicated web of changing interactions that are suggested by our current data.

Concerns/Suggestions:

1) Although this Reviewer understands why the authors intermingled the experiments using mammalian dynein and yeast dynein, I found the jumping back and forth between model systems very distracting. This reviewer views the role of anchored dynein (ie. yeast) as being a little different in mechanism and protein composition from motile dynein (ie. membrane transport). Needing to think about this for every experiment was challenging. In some ways the yeast dynein experiments and mammalian dynein experiments belong in separate papers. However, if the authors want to maintain the current style, I might recommend adding a Table (multi-column) in which they provide: 1) the mutant used, 2) the phenotype expected, and 3) some scoring scheme that indicates outcome. Lining up comparable mammalian and yeast constructs in the same rows would make interpretation easier.

Thank you for your comment and we agree that we have generated a large amount of data from both yeast and mammalian systems that could be challenging for some readers to parse through. However, we feel that the majority of the data are very consistent between the systems, so much so that it strongly supports a coherent model that is applicable to both systems. We also believe that, despite the minimal roles for dynein in yeast cells (*i.e.*, spindle positioning from the cell cortex), the biochemical mechanisms we describe here, along with prior work in the field, strongly suggest that the mechanisms governing yeast dynein activity are highly analogous to those in mammalian cells. Specifically, much like in human cells, dynein activity in yeast requires both dynactin and a coiled-coil-containing adapter protein (Num1; Lammers and Markus, JCB 2015). Both yeast and mammalian dynein systems are regulated by accumulation of components at microtubule plus-ends, and potential off-loading onto cargos such as the cell cortex in the case of yeast, or possibly vesicular cargo in mammalian systems. It's worth noting that human dynein also performs a spindle positioning function from the cell cortex (mediated by the presumed adaptor protein NuMA). Thus, in spite of ~1 billion years of evolutionary time between them, the mechanisms and regulators governing dynein activity in

yeast and humans appears to be highly similar. For these reasons, we feel our combined data from both model systems make a strong case for a unified model of regulation that will allow readers to see the extent of conservation between the two systems.

Although we appreciate the idea of a summary table, we were a bit unsure of what the reviewer had in mind, or what was meant by scoring phenotypes in some of our assays (e.g., binding assays). That being said, we have put together the table below to see if we hit the mark, or if the reviewer may have further suggestions for how to improve it for clarity. If the reviewer finds the table below useful, and thinks it will help to make our results more accessible to a broader audience, then we can include it as a supplementary item.

Table 1: Summary of expected phenotypes and experimental outcomes for mutants tested throughout study.

		dynein mutants		LIS1 mutants	Ndel1 mutants	
	mutant:	Δ ICN	IC ^{AAA}	R316A/W340A or "5A"	Ndel1 ^{AAA}	E119A/R130A or E64A/R78A
	expected outcome:	lack of DDA assembly/doesn't bind Ndel1	inability to bind Ndel1	inability to bind Ndel1	inability to bind IC/ability to bind LIS1	inability to bind LIS1/ability to bind IC
expected phenotype:	yeast	defects in spindle position, and localization of dynein and dynactin	no binding to Ndl1 in vitro	no binding to Ndl1 in vitro	no binding to IC in vitro/binding to Pac1	no binding to Pac1 in vitro/binding to dynein complex
	human	defects in spindle assembly and DDA assembly	no binding to Ndel1 in vitro	no binding to Ndel1 in vitro	no binding to IC in vitro/binding to LIS1	no binding to LIS1 in vitro/binding to IC
outcome validated?	yeast	yes (Fig. 2C, E and G)	yes (Fig. S3C)	yes (Fig. 4C)	yes (Fig. S3D and E)	yes (Fig. S3F and G)
	human	yes (Fig. 2A, B and D)	not tested	yes (Fig. 4D)	yes (Wang and Zheng, 2011 JBC)	yes (Fig. 5F; Wang and Zheng, 2011 JBC; Derewenda et al., 2007 Structure)

2) A number of important molecular transitions are suggested in the model proposed by the authors. These include Nde1 releasing LIS-1 for binding to the dynein motor domain, Nde1 releasing from the dynein IC N-terminal domain, binding of dynactin (P150Glued) to the same N-terminal domain of the dynein ICs, recruitment of the cargo adaptor (ie. Hook3) and release of LIS-1 by the dynein motor domain for motility. This sequential activation model is very interesting and compelling as a mechanism. However, the molecular basis of these transitions is not really explored to any extent. Simple diffusion-based equilibria are possible, but not very tangible. Given that the candidate proteins at the core of this model are all known to be phosphorylation substrates, it is surprising that the role of phosphorylation wasn't incorporated into the model. The molecular consequences of some of these phosphorylation events are known and phosphorylation sites have been mapped and mutated. The addition of mutants that affect phosphorylation status would be a powerful addition to the study. Especially if these mutants block transitions.

We agree with the idea that phosphorylation is likely a key mechanism controlling at least some aspects of the molecular transitions mentioned by the reviewer. In fact, we are currently working on experiments that test this idea in our labs. However, the results so far indicate that the role of phosphorylation is not straightforward, and that more time and effort will be needed to reveal a unified theory about the effects of phosphorylation on these transitions. One complicating factor

is that there are a large number of phosphorylation sites on these molecules, and an equally large repertoire of kinases responsible for the phosphorylation events. For example, a recently published study (Garrott and Gillies et al. JBC, 2023) explored the effects of some phosphorylation sites on Ndel1. They reported that phosphorylation of some sites within the C-terminus of Ndel1 lead to an increased affinity for dynein. In contrast to this phosphorylation event, another site on Ndel1 – T132 – has been proposed to reduce its affinity for LIS1 (Bradshaw et al. J. Neurosci, 2011). In principle, such a phosphorylation event that could aid in the “hand off” of LIS1 from Ndel1 to the dynein motor domain as proposed in our model. We have been examining the effects of this phosphorylation site on Ndel1, and our preliminary data indicate that a phospho-mimetic mutation at T132 indeed lowers the affinity of Ndel1 for LIS1, as reported. However, the result is not sufficiently strong to suggest a simple binary “on/off” switch that would explain the “hand off” of LIS1 to dynein in our model. Although we plan to continue to explore the effects of this phosphorylation site and others, this will require a more dedicated effort that is beyond the scope of the current paper. We have revised our discussion to more fully discuss a potential role of phosphorylation in the protein-protein interaction transitions we uncovered.

Reviewer #3 (Remarks to the Author):

Okada and Iyer et al. set out to understand the role that dynein intermediate chain plays in dynein activation. The authors explore this question with human and yeast systems using a combination of biochemistry, cell biology, and microscopy. The authors find that dynein’s ICD plays an important role in promoting the formation of dynein-dynactin-adaptor complexes. This finding is important because the role of IC-p150 binding in dynein activation is ill-defined since the regions of these proteins are not resolved in any published structures. The authors also find that Ndel1 antagonizes the formation of DDA complexes by binding competitively with p150 subunit of dynactin. Finally, the authors find that though Ndel1 binds competitively with dynein for Lis1 binding. This is a surprising finding that is in opposition to a well-established model of Lis1-Ndel1 function. These are important findings that address several important and outstanding questions. All the work was performed to a high standard and supports the claims that the authors make. I recommend this for publication.

Recommendations:

1. It is easy to get lost in the descriptions of interactions between Ndel1, dynein’s IC, and dynactin’s p150. Domain schematics for Ndel1 and p150 would be helpful in Figure 1 (as the authors have done for IC).

As suggested, we have now included protein schematics for p150 and Ndel1 in our revised manuscript (see Figs. 2H and 3A) to highlight the domains of the proteins that we work with in our experiments. We thank the reviewer for the suggestion.

2. I couldn’t find the experimental methods for the experiment in 2D and struggled to follow the experimental design. Can the authors add the methods for this assay? What data is plotted in the graph in 2D? Is that with or without added Hook3? Is it surprising that inclusion of Hook3 did not increase the apparent dynein-dynactin association with WT dynein?

We apologize for the omission of the methods for this experiment and have updated our methods section to describe the details. We have also renamed the Y-axis in the plot (Fig. 2D) to better describe the quantification, and updated the figure legend to describe the details of the

graph and quantitation. We agree it is somewhat surprising that the addition of Hook3 does not lead to a large increase in dynein-dynactin association. It was for this reason that we include an adapter-independent dynein-dynactin complex in our activation model (Fig. 6). We note there is abundant cellular evidence for such a complex, e.g., at microtubule plus-ends in yeast and mammalian systems, which we cite in our discussion: *“This adapter-independent dynein-dynactin-LIS1 complex may represent the plus-end bound dynein complexes observed in yeast and metazoans (Baumbach et al., 2017; Jha et al., 2017; Lee et al., 2003; Moore et al., 2008; Splinter et al., 2012).”* It is less clear to us why prior studies, including some from our own labs, have not found a robust biochemical interaction between isolated dynein and dynactin in the absence of a cargo adapter. We suspect it may be related to the novel method of isolation used in this study, which is pulldown via the strepII-tagged intermediate chain, but further experiments will be needed to confirm this.

3. The authors should show the Alphafold-generated ICD domain-docked on Ndel1 in the main text (some version of Figure S3A and B).

Our modified Figure 4A and B now include the Alphafold model of Ndel1 and Ndl1-bound ICN. We thank the reviewer for the suggestion.

4. Should cite Garrot and Gillies, JBC 2023 article to support the observations that Ndel1 disfavors DDA assembly and that Lis1 doesn't bind Ndel1 and dynein simultaneously.

We have updated our paper to discuss and cite Garrot and Gillies, JBC 2023, which makes nice complementary observations that further corroborate and support our data.

REVIEWERS' COMMENTS

Reviewer #1 (Remarks to the Author):

The authors have fully addressed the points I raised.
Minor point - Figure 5: the figure legend is missing.

Reviewer #2 (Remarks to the Author):

The responses submitted explaining why the authors could not address some suggestions I made are reasonable and acceptable.

The table was good starting point to help the readers. However, I took a stab at two other options for a explanatory table (attached). In thinking about what would help the reader most, I think linking the steps in the activation cycle to the specific constructs that reveal that step would be most useful (Version 2). It also reveals that there are some steps that were not tested in this study. If these steps were revealed in previous work, adding that would complete the table.

Otherwise I recommend acceptance.

Reviewer #3 (Remarks to the Author):

The authors addressed all reviewer points. I recommend the manuscript for publication.